# The clinical pharmacology of tafenoquine in the radical cure of *Plasmodium vivax* malaria: An individual patient data meta-analysis

James A Watson[1,2,3]*[†], Robert J Commons[3,4][†], Joel Tarning[2,5], Julie A Simpson[6], Alejandro Llanos Cuentas[7], Marcus VG Lacerda[8], Justin A Green[9], Gavin CKW Koh[10], Cindy S Chu[2,11], François H Nosten[2,11], Richard N Price[2,3,4], Nicholas PJ Day[2,5], Nicholas J White[2,5]*

[1]Oxford University Clinical Research Unit, Hospital for Tropical Diseases, Ho Chi Minh City, Viet Nam; [2]Centre for Tropical Medicine and Global Health, Nuffield Department of Medicine, University of Oxford, Oxford, United Kingdom; [3]WorldWide Antimalarial Resistance Network, Oxford, United Kingdom; [4]Global Health Division, Menzies School of Health Research, Charles Darwin University, Darwin, Australia; [5]Mahidol Oxford Tropical Medicine Research Unit, Faculty of Tropical Medicine, Mahidol University, Bangkok, Thailand; [6]Centre for Epidemiology and Biostatistics, Melbourne School of Population and Global Health, University of Melbourne, Melbourne, Australia; [7]Unit of Leishmaniasis and Malaria, Instituto de Medicina Tropical "Alexander von Humboldt", Universidad Peruana Cayetano Heredia, Lima, Peru; [8]Fundação de Medicina Tropical Dr Heitor Vieira Dourado, Manaus, Brazil; [9]Formerly Senior Director, Global Health, GlaxoSmithKline, Brentford, United Kingdom; [10]Department of Infectious Diseases, Northwick Park Hospital, Harrow, United Kingdom; [11]Shoklo Malaria Research Unit, Mahidol–Oxford Tropical Medicine Research Unit, Faculty of Tropical Medicine, Mahidol University, Mae Sot, Thailand

*For correspondence:
james@tropmedres.ac (JAW);
nickw@tropmedres.ac (NJW)

[†]These authors contributed equally to this work

**Abstract** Tafenoquine is a newly licensed antimalarial drug for the radical cure of *Plasmodium vivax* malaria. The mechanism of action and optimal dosing are uncertain. We pooled individual data from 1102 patients and 72 healthy volunteers studied in the pre-registration trials. We show that tafenoquine dose is the primary determinant of efficacy. Under an Emax model, we estimate the currently recommended 300 mg dose in a 60 kg adult (5 mg/kg) results in 70% of the maximal obtainable hypnozoiticidal effect. Increasing the dose to 7.5 mg/kg (i.e. 450 mg) would result in 90% reduction in the risk of *P. vivax* recurrence. After adjustment for dose, the tafenoquine terminal elimination half-life, and day 7 methaemoglobin concentration, but not the parent compound exposure, were also associated with recurrence. These results suggest that the production of oxidative metabolites is central to tafenoquine's hypnozoiticidal efficacy. Clinical trials of higher tafenoquine doses are needed to characterise their efficacy, safety and tolerability.

## Editor's evaluation

This competently performed retrospective analysis presents important findings concerning the clinical use of tafenoquine, a drug against Plasmodium vivax malaria. The assembly of the majority of global tafenoquine pharmacology data from clinical treatment studies provides compelling evidence in support of the drug's regimen that includes an increase in dosing, which would lead to a

significant enhancement of the drug efficacy, hence a decrease in recurrent parasitemia. The manuscript includes a detailed analysis and discussion concerning the side effects of the drug affecting more susceptible populations.

## Introduction

Tafenoquine is a newly licensed 8-aminoquinoline antimalarial drug for the radical treatment of relapsing *Plasmodium vivax* and *Plasmodium ovale* malarias; the first for 70 years. Tafenoquine is eliminated slowly and can be given as a single dose for radical cure or as weekly chemoprophylaxis (*Llanos-Cuentas et al., 2019*; *Lacerda et al., 2019*). This provides a substantial operational advantage over the currently recommended radical treatment, primaquine, which is eliminated rapidly and needs to be given at least once daily for 7–14 days (*Nekkab et al., 2021*; *Taylor et al., 2019a*). Primaquine is metabolised rapidly to carboxyprimaquine (considered pharmacologically inert) and, through a different pathway, to a series of reactive oxidative intermediates which are thought to be responsible for both antimalarial activity and haemolytic toxicity (*Camarda et al., 2019*; *White et al., 2022*). The main limitation in implementing the antimalarial 8-aminoquinolines is oxidative haemolysis. Haemolysis can be life-threatening in glucose-6-phosphate dehydrogenase (G6PD) deficiency, which is very common in malaria endemic areas (*Recht et al., 2014*). The optimum 8-aminoquinoline dose is therefore a balance between benefits (applicable to all treated patients) and risks (borne largely by the subgroup of patients with G6PD deficiency). Tafenoquine has been engineered to be much more stable than primaquine. The contribution of oxidative metabolites to the pharmacological activity of tafenoquine is unclear (*St Jean et al., 2020*). Indeed, the precise mechanism of action of the 8-aminoquinoline antimalarials in killing hypnozoites remains unknown.

Following a phase 2b dose-ranging trial, a single 300 mg dose of tafenoquine (approximately 5 mg base/kg) was chosen for phase 3 evaluation in adults (*Llanos-Cuentas et al., 2014*). The principal dose-limiting concern at the time was the haemolytic potential of tafenoquine in G6PD deficient heterozygote females (*Rueangweerayut et al., 2017*), who often test as 'normal' in G6PD deficiency rapid screens. The phase 3 tafenoquine registration trials overall were interpreted as showing approximate equivalence of 300 mg tafenoquine compared with lower dose primaquine (15 mg base daily for 14 days in adults), although the pre-specified non-inferiority margin was not met, and in Southeast Asia, tafenoquine efficacy was significantly inferior to primaquine (*Llanos-Cuentas et al., 2019*). *P. vivax* parasites from Southeast Asia and Oceania are considered to require higher 8-aminoquinoline doses for radical cure than elsewhere. The World Health Organization recommends for adults a total primaquine dose of 420 mg base (30 mg daily) in these regions compared with 210 mg (15 mg daily) in other endemic regions (*World Health Organization, 2015*). In the Southeast Asian region, the 300 mg adult dose of tafenoquine was inferior to a sub-optimal primaquine radical cure regimen, although the sample size was small (43 patients were randomised to low-dose primaquine, and 73 were randomised to tafenoquine 300 mg). In the recently reported INSPECTOR trial, 300 mg single dose tafenoquine resulted in poor radical curative efficacy in Indonesian soldiers returning from Papua, Indonesia, an area of high malaria transmission (*Baird et al., 2020*). Several hypotheses for these disappointing results have been proposed including underdosing (*Watson et al., 2021*) and an antagonistic interaction between dihydroartemisinin-piperaquine (DHA-PQP) and tafenoquine. Although a single dose tafenoquine recommendation for all adults has practical advantages, because of variation in body weight, the single dose results in substantial variations in drug exposure. Taken together, it seems likely that the currently recommended adult dose of tafenoquine 300 mg is inferior to optimal primaquine regimens in preventing relapses of vivax malaria in all endemic regions (*White, 2021*; *Watson et al., 2021*).

The 8-aminoquinoline drugs are considered to be pro-drugs, although there has been some debate about this for tafenoquine (*St Jean et al., 2020*). For primaquine, plasma concentrations of the parent compound and the inactive carboxy metabolite can be measured readily, but characterisation of the active metabolites, or their biotransformed derivatives, has proven challenging. Tafenoquine has a terminal elimination half-life of about 15 days compared with 5 hours for primaquine, so capturing transient active metabolites is even more difficult. The current inability to measure the biologically active moiety impedes pharmacometric dose optimisation trials. We have suggested that increases in red cell methaemoglobin (MetHb) production following 8-aminoquinoline administration could be

**Table 1.** Demographic, clinical and dosing information on tafenoquine and primaquine dosing groups for all patients included in the efficacy analyses.

(*n* = 1073, flow diagram shown in *Appendix 1—figure 1*). Unless otherwise stated, for binary variables we show the total number (%); for continuous variables we show the mean (standard deviation). * Median (interquartile range).

| | No TQ | PQ | TQ <3.75 mg/kg | TQ [3.75,6.25] mg/kg | TQ [6.25,8.75) mg/kg | TQ≥8.75 mg/kg |
|---|---|---|---|---|---|---|
| | *n*=182 | *n*=257 | *n*=169 | *n*=368 | *n*=54 | *n*=43 |
| Recurrence at 4 months | 101 (55.5) | 57 (22.2) | 57 (33.7) | 79 (21.5) | 4 (7.4) | 0 (0) |
| Age (years) | 35 (14.1) | 36 (14.4) | 37 (13.4) | 36 (14.2) | 36 (17.2) | 33 (14.4) |
| Female | 50 (27) | 77 (30) | 43 (25) | 95 (26) | 24 (44) | 9 (21) |
| Weight (kg) | 62 (12.0) | 63 (12.1) | 72 (20.1) | 63 (8.7) | 51 (13.3) | 57 (7.3) |
| Baseline parasitaemia (/μL)* | 5470 (2173–11856) | 4697 (1712–10430) | 4320 (1447–9456) | 4174 (1431–10101) | 5507 (1961–12111) | 6143 (1692–13313) |
| Haemoglobin day 0 (g/dL) | 12.9 (1.5) | 13.0 (1.6) | 12.9 (1.8) | 13.2 (1.6) | 12.3 (1.8) | 12.6 (1.8) |
| *Country* | | | | | | |
| Brazil | 55 (30) | 80 (31) | 61 (36) | 105 (29) | 6 (11) | 3 (7) |
| Cambodia | 10 (5) | 8 (3) | 0 (0) | 16 (4) | 2 (4) | 0 (0) |
| Colombia | 0 (0) | 5 (2) | 0 (0) | 11 (3) | 0 (0) | 0 (0) |
| Ethiopia | 14 (8) | 13 (5) | 0 (0) | 18 (5) | 7 (13) | 0 (0) |
| India | 9 (5) | 5 (2) | 21 (12) | 8 (2) | 4 (7) | 7 (16) |
| Peru | 62 (34) | 91 (35) | 56 (33) | 136 (37) | 21 (39) | 18 (42) |
| Philippines | 1 (1) | 2 (1) | 0 (0) | 3 (1) | 0 (0) | 0 (0) |
| Thailand | 31 (17) | 38 (15) | 31 (18) | 46 (12) | 10 (19) | 15 (35) |
| Vietnam | 0 (0) | 15 (6) | 0 (0) | 25 (7) | 4 (7) | 0 (0) |

used as an in vivo pharmacodynamic proxy of oxidative antimalarial activity (*White et al., 2022*). MetHb is an oxidation product in which the ferrous iron ($Fe^{2+}$) of intraerythrocytic haem has been oxidised to the ferric form ($Fe^{3+}$). Data from over 500 *P. vivax* malaria patients given high-dose primaquine radical cure suggested that increases in day 7 blood methaemoglobin concentrations (which is approximately the day of peak methaemoglobinaemia in a daily primaquine radical cure regimen) were associated with greater radical curative efficacy (*Chu et al., 2021*).

To guide optimal tafenoquine dosing and to identify factors associated with radical cure efficacy and haemolysis, we analysed the relationship between tafenoquine dose, pharmacokinetics, MetHb production, haemoglobin reduction and recurrence of vivax malaria using data from the three tafenoquine registration trials conducted in patients with acute vivax malaria (DETECTIVE phase 2b, DETECTIVE phase 3, and GATHER) (*Llanos-Cuentas et al., 2014*; *Lacerda et al., 2019*; *Llanos-Cuentas et al., 2019*). Rich sampling data from a phase 1 study of tafenoquine pharmacokinetics in healthy volunteers (*Green et al., 2016*) were pooled with the patient data to develop a population pharmacokinetic model of tafenoquine, and thus characterise individual patient tafenoquine exposure and elimination. Our results provide important insights into the pharmacodynamics of tafenoquine, notably the central importance of oxidative metabolites. From this we show that the currently recommended dose of tafenoquine has sub-optimal efficacy, and we propose evaluation of a new 50% higher dose (450 mg adult dose) that could reduce vivax malaria recurrences substantially with little predicted increase in the risk of severe adverse events.

## Results

### Weight-adjusted radical cure efficacy of single-dose tafenoquine

Efficacy data were pooled from 1073 patients with acute *P. vivax* malaria who were recruited in nine countries (*Table 1*) between October 2010 and May 2017 (*Appendix 1—figure 1*). All patients were treated with a standard dose regimen of chloroquine (1500 mg base equivalent over 3 days) and were

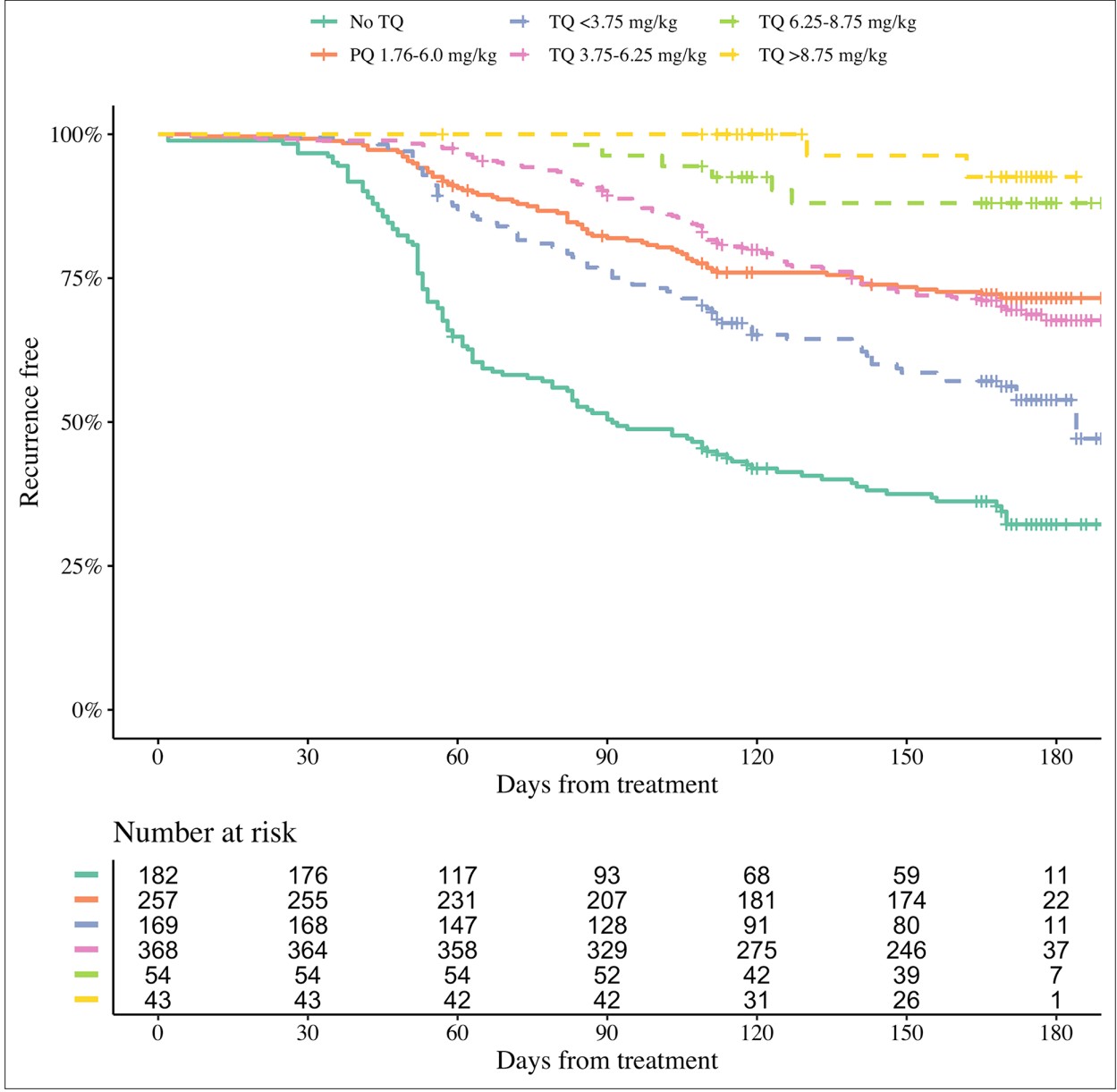

**Figure 1.** Kaplan-Meier survival curves for the time to first recurrence in 1073 patients with *P. vivax* malaria in the efficacy analysis population. Patients are grouped by tafenoquine mg/kg dosing category (dashed and dotted lines) versus low-dose primaquine (15 mg base for 14 days) or no radical cure (thick lines). The vertical ticks show the censoring times.

randomised to either single dose tafenoquine (varying doses), low-dose primaquine (15 mg base equivalent daily for 14 days), or to no 8-aminoquinoline treatment. In the patients who received tafenoquine (*n*=634), single doses ranged from 50 mg to 600 mg, with weight adjusted doses ranging from 0.5 mg base/kg to 14.3 mg/kg (i.e. a greater than 25-fold variation of weight adjusted dose). *Figure 1* shows the Kaplan-Meier survival curves for the pooled data on time to first recurrence, grouped by tafenoquine dosing category, and compared with low-dose primaquine (i.e. adult dose 15 mg/day for 14 days) and no radical cure regimen. Patient weight (and thus mg/kg tafenoquine dose) varied considerably across the nine countries, with the heaviest patients in Brazil (median weight 70 kg) and the lightest patients in Ethiopia and the Philippines (median weight 53 kg), see *Appendix 1—figure 2*.

We first characterised the association between the mg/kg tafenoquine dose and the odds of recurrence of *P. vivax* malaria using logistic regression. Under the model fit to all data (*n*=1073), each additional mg/kg of tafenoquine was associated with an odds-ratio (OR) for recurrence within

4 months of 0.70 (95% CI: 0.65–0.76). A 1 mg/kg tafenoquine increase corresponds to a 20% increase relative to the currently recommended dose of 5 mg/kg (assuming a patient weight of 60 kg). In comparison, a 20% increase in the total primaquine dose corresponds to 0.7 mg/kg for a target dose of 3.5 mg/kg. A 0.7 mg/kg increase in the total primaquine dose was associated with an odds ratio for recurrence within 4 months of 0.72 (95% CI 0.66–0.78): a very similar dose effect for the two drugs. Our model estimated that the mean probability of recurrence at 4 months was 0.15 (95% CI: 0.10–0.23) following 5 mg/kg tafenoquine and 0.17 (95% CI: 0.10–0.26) following 3.5 mg/kg primaquine. The posterior probability that the recurrence was more likely following primaquine was 0.73. The effect of tafenoquine mg/kg dose was similar in an analysis restricted to patients who received the currently recommended 300 mg dose ($n$=469, OR = 0.66; 95% CI 0.51–0.85). The association between tafenoquine mg/kg dose and recurrence was observed in all three geographic

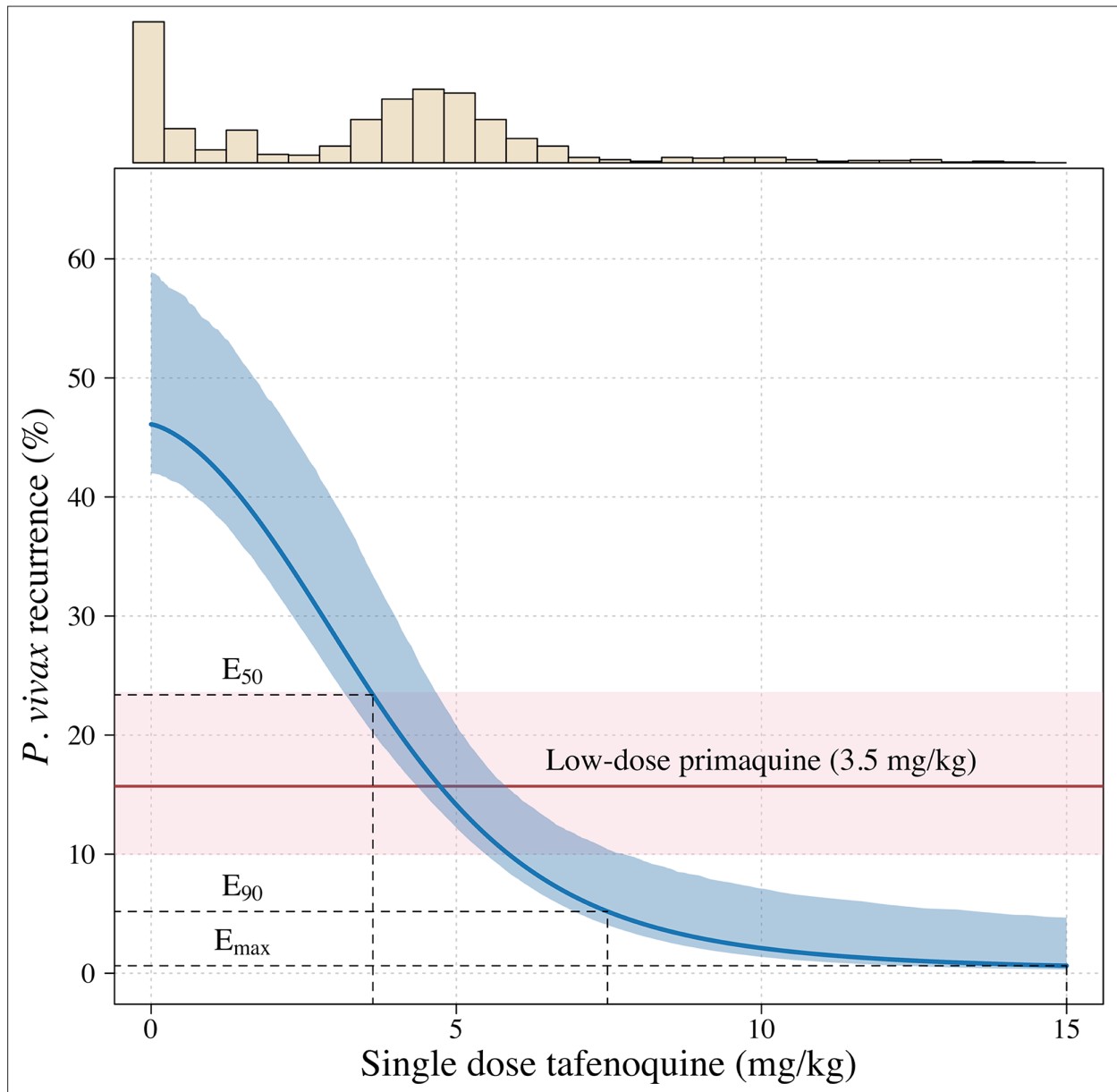

**Figure 2.** Tafenoquine mg/kg dose and the probability of a recurrence of *P. vivax* malaria within 4 months under the Emax model. Thick blue (shaded blue): mean probability of recurrence as a function of the tafenoquine mg/kg dose (95% CI); thick red (shaded pink): mean probability of *P. vivax* recurrence after 3.5 mg/kg primaquine (95% CI). The distribution of administered tafenoquine mg/kg doses is shown by the histogram above (this includes patients who received no radical cure treatment, but for clarity does not include patients who received primaquine: $n$=816; range: 0–14.3 mg/kg).

regions (Americas, Horn of Africa and Southeast Asia: *Appendix 1—figure 3*), with near identical results when using any recurrence by 6 months as the endpoint, or analysing by time to first recurrent event (*Appendix 1—figures 4 and 5*).

Further analysis under an Emax model indicated that the mg/kg dose-recurrence relationship was not linear on the log-odds scale. No recurrences by 4 months were observed in any patients in the highest dosing group (≥8.75 mg/kg, *n* = 43). The Emax model estimated a slightly steeper dose-response curve compared to logistic regression fits (*Appendix 1—figure 6*). *Figure 2* shows the estimated dose-response curve for recurrence at 4 months under the Emax model. Taking 15 mg/kg as the maximum tolerated single dose of tafenoquine, a dose of approximately 7.5 mg/kg achieves 90% of the effect achieved with 15 mg/kg (*Appendix 1—figure 7*). Near identical results were obtained for the 6 month endpoint (*Appendix 1—figure 8*). Under the Emax model, the dose-response is steep around doses of approximately 2.5mg/kg to 5 mg/kg. For example, increasing the dose from 3mg/kg to 4 mg/kg reduces the mean estimated recurrence proportion from 28.5% to 20.8%; increasing to 5 mg/kg gives a mean recurrence proportion of 14.2%.

## Predicted efficacy of higher dose tafenoquine

Using the empirical weight distribution from the three efficacy trials, we estimate that a fixed 300 mg dose in an adult would result in a mean recurrence proportion of 15.3% (95% CI: 9.6–21.7), whereas a 450 mg fixed dose would result in a mean recurrence proportion of 6.2% (95% CI: 3.2–11.5). Given that approximately half the patients with no 8-aminoquinoline treatment had a recurrence, this implies that the 300 mg fixed dose reduces recurrence rates by around 70%, whereas a 450 mg fixed dose would reduce them by around 85%. *Appendix 1—figure 9* shows the expected distributions of individual efficacy (relative to all patients receiving 15 mg/kg) for the 300 mg versus 450 mg fixed dose, based on the empirical distribution of weights in the three efficacy studies. This illustrates that small dose increments can result in substantial increases in radical cure efficacy. If the 300 mg adult dose was increased to 450 mg this would translate into a number needed to treat of 11 patients (95% CI 7–21) to prevent one additional recurrence of malaria (*Appendix 1—figure 10*) with the greatest absolute benefit for patients weighing over 60 kg (*Appendix 1—figure 11*).

## Pharmacokinetic predictors of tafenoquine radical cure efficacy

We fitted mixed-effects population pharmacokinetic models to all available plasma tafenoquine levels (a total of 4499 tafenoquine plasma concentrations in 718 individuals: 72 healthy volunteers and 646 patients with symptomatic *P. vivax* malaria). The data were described adequately by a two-compartment elimination model, consistent with previous analyses of these data (*Thakkar et al., 2018*). For each patient who received tafenoquine and who had at least one measurable plasma level recorded (n=646), drug exposures and metabolism were summarised by the total area under the plasma concentration time curve ($\rho$), the maximum plasma concentration ($C_{max}$), and the terminal elimination half-life ($t_{1/2}$).

Higher plasma tafenoquine $AUC_{[0,\infty)}$ values were associated with lower odds of recurrence at 4 months (OR: 0.61 [95% CI 0.47–0.77] for each standard deviation increase in $AUC_{[0,\infty)}$). However, this relationship was driven primarily by dose, that is the differences in $AUC_{[0,\infty)}$ between those who received lower (50 mg or 100 mg) versus higher (300 mg or 600 mg) doses (*Appendix 1—figure 12*). In a model adjusted for tafenoquine mg/kg dose, the $AUC_{[0,\infty)}$ was no longer associated with recurrence at 4 months (OR: 0.89 per standard deviation change in $AUC_{[0,\infty)}$, 95% CI 0.64–1.23). The same pattern was seen for the $C_{max}$: in a univariable model there was a strong association between $C_{max}$ and recurrence, but in a model adjusted for tafenoquine mg/kg dose this association was no longer significant (OR: 1.01 for a doubling in $C_{max}$, 95% CI 0.69–1.49). Thus dose administered rather than parent compound exposure was the main observed driver of therapeutic efficacy.

Assuming linear kinetics, the weight-adjusted $t_{1/2}$ is, by definition, independent of the mg/kg dose. In contrast to exposure data ($AUC_{[0,\infty)}$ and $C_{max}$), the individual tafenoquine weight-adjusted $t_{1/2}$ estimates were associated with recurrence at 4 months. More rapid tafenoquine elimination (despite the lower parent drug exposure) was associated with lower odds of recurrence. In a model adjusted for tafenoquine mg/kg dose, the odds ratio for recurrence at 4 months for each day increase in $t_{1/2}$ was 1.15 (95% CI 1.06–1.25).

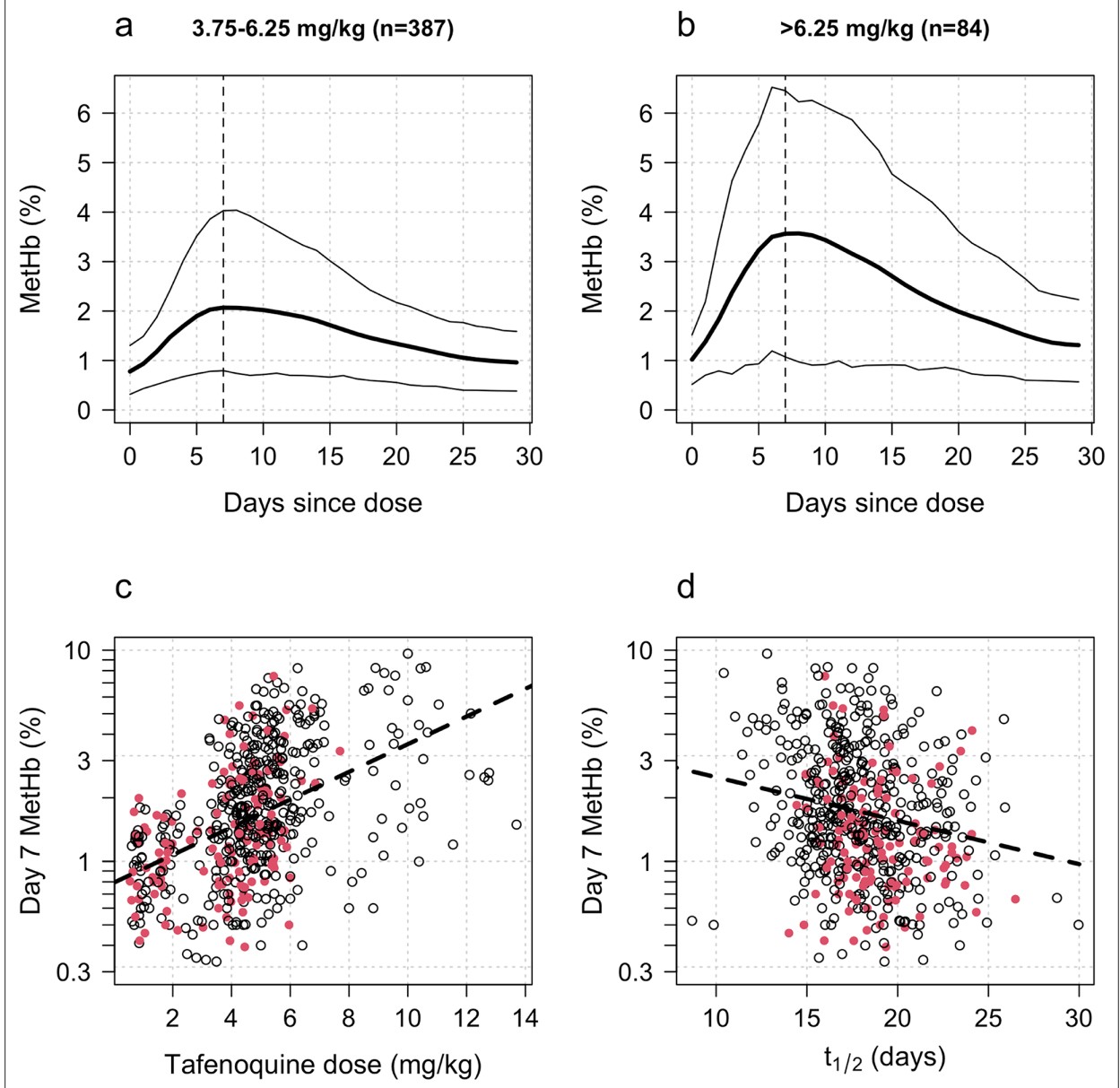

**Figure 3.** Tafenoquine and methaemoglobin (MetHb) production. Panels a and b: dose-dependent increases in blood MetHb concentrations expressed as a proportion (%) of the haemoglobin concentration following tafenoquine +chloroquine administration in patients with *P. vivax* malaria (mean: thick lines; 10th and 90th percentiles: thin lines). The approximate peak concentrations occur one week after receiving tafenoquine (shown by the vertical dashed lines). Panel c: relationship between the administered tafenoquine doses (mg/kg) and the interpolated MetHb concentrations on day 7 ($log_{10}$ scale). Panel d: relationship between the estimated individual tafenoquine terminal elimination half-lives ($t_{1/2}$) and the interpolated MetHb concentrations on day 7 ($log_{10}$ scale). Panels c and d: patients who had a recurrence within 4 months are shown by the filled red circles.

## Methaemoglobin production and tafenoquine radical cure efficacy

Increases in blood MetHb concentrations following tafenoquine administration were highly correlated with the mg/kg dose (*Figure 3a–c*). Each additional mg/kg was associated with a 19% (95% CI: 17% to 21%) increase in day 7 MetHb concentrations. As reported previously for high-dose primaquine (*Chu et al., 2021*), peak MetHb levels were observed around day 7 (*Figure 3a–b*). Adding the day 7 MetHb concentration as a linear predictor in the logit-Emax model (i.e. in a model adjusted for the mg/kg dose of tafenoquine), each absolute percentage point increase in the day 7 MetHb concentration was associated with an odds ratio for recurrence of 0.81 (95% CI 0.65 to 0.99). Furthermore, consistent with this association between MetHb and vivax malaria recurrence (*Figure 4*), the day 7 MetHb levels

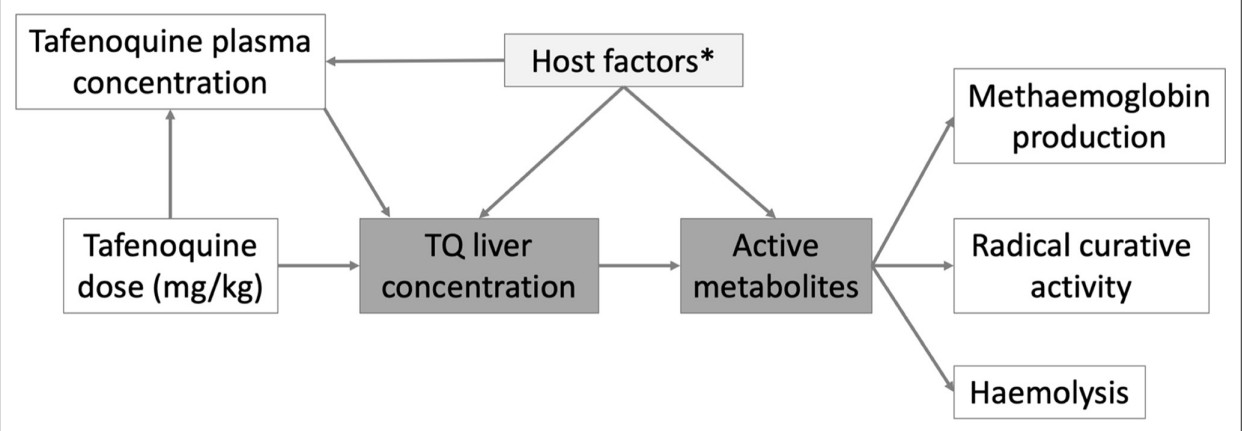

**Figure 4.** Hypothetical causal model of the clinical pharmacology of tafenoquine for the radical treatment of *P. vivax* malaria. The dark grey shaded boxes represent non-observables; lighter shaded grey represents partially observable measures. *Potential host factors include genetic factors such as *CYP2D6* polymorphisms; age related enzyme maturation; drug-drug interactions.

were weakly inversely correlated with the tafenoquine weight-adjusted $t_{1/2}$ values (*Figure 3d*; $\rho$ =-0.1, *P*=0.05).

The high *P. vivax* recurrence rates following tafenoquine plus DHA-PQP in the INSPECTOR study (*Baird et al., 2020*) raised concerns over potential drug-drug interactions between tafenoquine and partner schizontocidal drugs. We compared day 7 MetHb concentrations in patients and healthy volunteers as a function of the partner antimalarial drug (all *P. vivax* patients were given chloroquine, *n*=578; healthy volunteers received either no partner drug [*n*=24], DHA-PQP [*n*=24], or artemether-lumefantrine (AL) [*n*=24]), adjusting for the tafenoquine mg/kg dose (the major predictor of day 7 MetHb). Overall, after adjustment for the mg/kg dose, healthy volunteers had slightly lower day 7 methaemoglobinaemia compared to patients infected with *P. vivax* (15% lower, 95% CI: 0 to 34% p=0.02), *Appendix 1—figure 13*. When comparing by randomised arm in the healthy volunteer trial, the largest observed difference was in those who received DHA-PQP in whom the day 7 MetHb was 23% lower (95% CI: 1.5 to 40%) compared with those volunteers who were not given a partner drug. However, it is not possible to differentiate between drug effects and disease effects when comparing malaria patients with healthy volunteers.

Lumefantrine may inhibit cytochrome P450 2D6 (CYP2D6) (*Novartis Pharmaceuticals, 2021*, see next section). We compared day 7 lumefantrine concentrations and day 7 piperaquine concentrations with the day 7 MetHb concentration in individuals randomised to tafenoquine + ACT in the phase 1 trial (*Green et al., 2016*). Higher plasma lumefantrine concentrations were associated with lower day 7 MetHb levels (ten-fold increase in day 7 plasma lumefantrine concentration was associated with a 40% reduction in day 7 methaemoglobin (95% CI: 3 to 62%), *Appendix 1—figure 14*). In contrast, for piperaquine there was no clear association with the day 7 MetHb concentrations (40%; 95% CI: –27to 72%).

### *CYP2D6* polymorphisms

CYP2D6 metabolises primaquine to a series of bioactive hydroxylated metabolites (*Camarda et al., 2019*). Genetic polymorphisms in *CYP2D6* resulting in loss of function are associated with reduced primaquine radical curative efficacy (*Baird et al., 2018*; *Baird, 2019*). A total of 716 patients had *CYP2D6* genotyping data, with a wide range of variants observed. The most common poor metaboliser alleles were *10 (10.5% allele frequency), *4 (5.8% allele frequency) and *5 (3.1% allele frequency). In the efficacy analysis population, of those randomised to tafenoquine, 34 (8.9%) patients had a predicted enzyme activity score of 0.5 or less (poor metabolisers) (*Caudle et al., 2020*). 67% (23/34) of these patients were enrolled in Southeast Asia where *10 has high allele frequencies. Consistent with the previous analysis of these data (*St Jean et al., 2016*) there was no clear association between the 2D6 activity score and vivax malaria recurrence at 4 months following tafenoquine (OR for recurrence: 0.9 [95% CI 0.4to 2.2] for poor versus normal/extensive metabolisers), although statistical power was low as few patients had low activity scores. In addition, there was no clear association between having

an activity score ≤0.5 ($n$=35 vs $n$=356) and tafenoquine elimination half-life (–0.15 days change in the poor versus normal/extensive metabolisers, 95% CI: –1.22 to 0.94).

## Tolerability and safety

In total 2.3% (15/651) of patients vomited within 1 hour of oral tafenoquine administration. There was no clear association with the mg/kg dose (OR for vomiting per mg/kg increase: 1.08 [95% CI: 0.88–1.32]). Severe haemolytic events were rare. One patient treated with tafenoquine had a greater than 5 g/dL fall in Hb, but no patients had a greater than 25% fall in Hb to <7 g/dL, or a fall to less than 5 g/dL. In a mixed effects linear regression model each additional mg/kg of tafenoquine was associated with a haemoglobin change at day 2 or 3 of –0.02 g/dL (95% CI: –0.04 to 0.00; $P$ = 0.02). This implies that a 300 mg tafenoquine dose in a 60 kg adult would result in an additional mean haemoglobin reduction at day 2 or 3 of around 0.1 g/dL (i.e. <1% reduction in Hb). Increasing the dose to 450 mg would result in a 0.15 g/dL mean reduction. In a separate model, day 7 MetHb concentration was correlated with haemoglobin reductions on day 2 or 3 (as postulated in the causal diagram in *Figure 4*). A one percent increase in MetHb was associated with a change in haemoglobin concentration of –0.08 g/dL (95% CI: –0.12 to –0.03). In contrast, every one day increase in the tafenoquine terminal elimination half-life was associated with a haemoglobin increase of 0.02 g/dL (95% CI: 0.01 to 0.04).

## Discussion

Our analysis provides strong evidence that the currently recommended adult dose of tafenoquine is insufficient for radical cure in all adults. In endemic areas, relapse of vivax malaria is a major contributor to morbidity and, in areas of high transmission, a major contributor to mortality - particularly in young children (*Douglas et al., 2012*; *Commons et al., 2019*; *Dini et al., 2020*). Tafenoquine can prevent vivax malaria relapses in a single dose treatment. It is potentially, therefore, a major advance in antimalarial therapeutics (*Nekkab et al., 2021*). But suboptimal dosing markedly reduces its therapeutic utility. Our pharmacometric assessment shows that the tafenoquine antimalarial dose-response relationship is steep around the currently recommended dose. Heavier patients are particularly disadvantaged. Substantial improvements in radical curative efficacy are possible with small dose increases. The tafenoquine dose-response relationship for haemolysis in G6PD deficiency is much less well characterised (*Rueangweerayut et al., 2017*), although the currently recommended dose is considered potentially dangerous in all G6PD deficient individuals, including female heterozygotes who may still have a large proportion of G6PD deficient red cells. For these reasons substantial efforts have been made to ensure accurate quantitative testing so that tafenoquine is not given to G6PD deficient individuals. The efficacy, tolerability and safety of increased doses should now be evaluated in prospective studies. Given the substantial investment to date in developing and promoting tafenoquine, and the associated investments in developing field applicable methodologies for quantitative G6PD estimation, a relatively small further investment to find the correct dose would seem thoroughly worthwhile. Assuming a distribution of body weights similar to that in the tafenoquine pre-registration trials, we estimate that increasing the current dosing recommendation from 300mg to 450mg will more than halve recurrence rates at 4 months. This would be operationally feasible given the current 150 mg tablet size. A 50% increase in the adult dose, resulting in at most 12.9 mg/kg in a 35 kg adult, is predicted to avert one relapse of malaria for every 11 patients treated. Tafenoquine is generally well tolerated. Much larger doses have been studied in treatment and prophylaxis trials (up to 2100 mg given over one week, *Walsh et al., 1999*, see Supplementary Appendix). The only report of a severe haemolytic reaction occurred in a female patient who was heterozygous for G6PD deficiency (A-variant) and received a total dose of 1200 mg tafenoquine over 3 days (*Shanks et al., 2001*). In the same study, a homozygous female (A- variant) who was also given 1200 mg tafenoquine over 3 days had an estimated 3 g/dL drop in haemoglobin, but remained asymptomatic.

The mechanism of the relapse preventing (radical curative) action of tafenoquine is still uncertain. This large individual patient data meta-analysis, which comprises data from the majority of all patients ever enrolled in tafenoquine clinical trials, shows clearly that the radical curative activity of tafenoquine in vivax malaria results from the activity of its oxidative metabolites. Dose, rather than exposure to the parent compound, was the main determinant of radical curative efficacy. Despite reducing total

exposure, more rapid tafenoquine elimination, and therefore greater biotransformation was associated with increased efficacy. This suggests, unsurprisingly, that tafenoquine behaves pharmacologically like a slowly eliminated form of primaquine (*Gonçalves et al., 2017*). The central importance of the oxidative metabolites in antimalarial activity was confirmed in these large and detailed studies by the relationship of radical curative efficacy with methaemoglobinaemia. A similar relationship has been shown for other 8-aminoquinolines (*Chu et al., 2021*; *White et al., 2022*). The reversible oxidation of intraerythrocytic haemoglobin reflects oxidative activity, and in this large series was correlated positively with relapse prevention and negatively with the tafenoquine half-life. Tafenoquine causes very minor oxidative haemolysis in G6PD normal individuals, and this too correlated with methaemoglobinaemia. These findings are all internally consistent with the hypothesis that oxidative activity is necessary for radical curative activity and also mediates haemolysis. Although tafenoquine has been engineered to be stable, it therefore does require metabolism for its pharmacological activity. For primaquine the unstable hydroxylated metabolites have been implicated as the active moieties, with CYP2D6 playing an important role in their formation (*Pybus et al., 2012*). The identity of the tafenoquine active oxidative metabolites, and the metabolic pathway have not been characterised yet. These data confirm the value of methaemoglobinaemia as a readily quantifiable non-invasive pharmacodynamic correlate of the antimalarial activity of the 8-aminoquinoline antimalarials (*White et al., 2022*; *Chu et al., 2021*).

The inferred steep antimalarial dose-response relationship provides a satisfactory explanation for the poor recent results with tafenoquine reported in the INSPECTOR trial (*Baird et al., 2020*) from Indonesia. In this trial, the currently recommended fixed dose of 300 mg was given to a group of Indonesian soldiers who had been exposed to high malaria transmission. The soldiers were heavier than the Southeast Asian patients studied in the registration trials, with a mean body weight of 70 kg. As Southeast Asian *P. vivax* infections are thought to require larger doses of primaquine than elsewhere, it is therefore likely that the high preceding transmission in Papua (heavier hypnozoite burdens, *White, 2021*) and lower mg/kg dosing contributed substantially to the very poor radical curative efficacy.

## Limitations

Our analysis was confined to assessment of the determinants of radical curative efficacy. The two major limitations of the study are the absence of direct measurements of the bioactive metabolites of tafenoquine, and imprecision in the measurement of relapse. As relapses can be with parasites which are genetically unrelated to the primary infection, distinguishing recrudescence, relapse and reinfection is difficult. Combining genotyping with time to event analysis improves discrimination (*Taylor et al., 2019b*). In these clinical trials, parasite genotyping was only done at three microsatellite loci and raw genotyping data could not be obtained to infer probabilistic assessments of relapse versus reinfection (*Taylor et al., 2019b*). However, these studies were conducted in generally low transmission settings, and over 4 out of 5 recurrences occurred within four months of treatment which makes a significant contribution from reinfection unlikely (*Figure 1*). We note that the similar results of 4 month and 6 month efficacy endpoints suggests that future antirelapse efficacy studies could be restricted to a follow-up of four months. This has substantial implications for the cost and operational feasibility of future antirelapse studies. Not only did most observed recurrences occurred within 4 months follow-up (83%, 298 out of 360), but this endpoint also had considerably less loss to follow-up.

As tafenoquine metabolites could not be measured their production was inferred from the parent drug elimination rate. The pharmacodynamic properties of the oxidative metabolites were inferred from measurements of methaemoglobin blood concentrations. Methaemoglobin production does not lie on the causal pathway to the killing of hypnozoites, and measurements show substantial variation between patients, but it proved to be a useful, non-invasive, easily measured, and therefore readily deployable surrogate for radical curative activity. The observed correlation of recurrence with terminal elimination half-life and MetHb production strongly supports the proposed causal model (*Figure 4*). Although the importance of drug metabolism was shown, our large study was insufficiently powered to evaluate the role in tafenoquine bioactivation of *CYP2D6* genetic polymorphisms (which have been shown to be important in bioactivating primaquine). Further work to characterise tafenoquine metabolism is needed.

Only DETECTIVE part 1 randomised patients to different tafenoquine doses; in the other two studies, variation in mg/kg dose results from variation in patient weight. Therefore, confounding

because of population differences in recurrence risk for patients of different body weight (e.g. different socioeconomic status) cannot be excluded. However, patient weight was not associated with recurrence. Over half the variation in body weight is polygenic, which is highly unlikely to be related to malaria risk (*Maes et al., 1997*). Importantly, the relationship between dose and recurrence was consistent across all three geographic regions. The results are nearly identical when only analysing data from DETECTIVE part 1 (OR: 0.62 per mg/kg increase [95% CI 0.52–0.72]).

A final limitation concerns the role of *CYP2D6* human genetic polymorphisms in the metabolism of primaquine. Only two thirds of enrolled patients were genotyped for *CYP2D6* mutations. In patients who received tafenoquine, only 35 were poor metabolisers. This makes it difficult to say with certainty what role CYP2D6 has in mediating tafenoquine metabolism. However, it highlights the fact that in these vivax malaria endemic areas, even if *CYP2D6* polymorphisms did play a key role, relatively few patients are null metabolisers. Thus CYP2D6 is a minor determinant of efficacy at the population level compared with drug dosing.

## Conclusions

These well conducted, detailed pre-registration clinical studies provide a clear description of the clinical pharmacology of this new slowly eliminated 8-aminoquinoline in the radical cure of vivax malaria. The results show the central importance of tafenoquine metabolism to oxidative intermediates in determining therapeutic efficacy. They allow accurate characterisation of the tafenoquine antimalarial dose-response relationship, and thereby provide compelling evidence that the currently recommended adult dose is insufficient. In order to optimise the radical cure of vivax malaria with tafenoquine, prospective studies should now be instituted to assess the tolerability, safety and efficacy of higher doses. Increasing the adult dose to 450 mg is predicted to reduce the risk of relapse substantially, particularly in heavier patients.

## Materials and methods

This study was a meta-analysis of individual volunteer and patient data gathered during the detailed pre-registration studies of tafenoquine. These comprise the majority of all tafenoquine clinical trial data. The flow diagram in *Appendix 1—figure 1* shows how the different analysis data sets were constructed. The results of all four studies included have been published previously (*Green et al., 2016*; *Llanos-Cuentas et al., 2014*; *Lacerda et al., 2019*; *Llanos-Cuentas et al., 2019*). Our analyses include 77% (651/847) of all patients given tafenoquine in vivax malaria treatment studies (see Supplementary Materials). Notable missing data are from the TEACH trial (single arm paediatric trial of weight based tafenoquine doses in symptomatic *P. vivax*, *Vélez et al., 2022*) and the INSPECTOR trial (returning soldiers, *Baird et al., 2020*) as the individual patient data from these two studies are not yet available to independent researchers.

### Ethical review

Anonymised individual patient data were obtained via ClinicalStudyDataRequest.com following approval of a research proposal from the Independent Review Panel. Re-use of existing, appropriately anonymised, human data does not require ethical approval under the Oxford Tropical Research Ethics Committee regulations (OxTREC).

### Data

In all studies, healthy volunteers and patients were required to give written, informed consent. All patients and volunteers were screened for G6PD deficiency. The patients were non-pregnant adults with uncomplicated *P. vivax* malaria and greater than 70% of the population median G6PD enzyme activity. The studies were approved by local review boards/ethics committees and were conducted in accordance with the Declaration of Helsinki and Good Clinical Practice guidelines.

We pooled clinical efficacy, laboratory, and pharmacokinetic data from four studies in adults: the phase 2b dose-ranging trial DETECTIVE part 1 (NCT01376167, *Llanos-Cuentas et al., 2014*); the phase 3 confirmatory trial DETECTIVE part 2 (NCT01376167, *Lacerda et al., 2019*); the phase 3 safety trial GATHER (NCT02216123, *Llanos-Cuentas et al., 2019*); and a phase 1 pharmacokinetic study

which assessed drug-drug interactions in healthy volunteers (NCT02184637, laboratory and pharmacokinetic data only, *Green et al., 2016*).

## Drug administration

In all studies tafenoquine was administered orally as a single dose. In the DETECTIVE and GATHER trials, tafenoquine was administered in combination with the standard chloroquine malaria treatment regimen for adults (1500 mg base equivalent given over 2–3 days). In these trials primaquine, 15 mg (base equivalent) daily given for 14 days, was the comparator. The phase 1 study was a drug-drug interaction study, with volunteers randomised to either tafenoquine alone or tafenoquine in combination with DHA-PQP or artemether-lumefantrine (AL), the standard antimalarial treatment regimens in both cases. In DETECTIVE part 1, tafenoquine was administered as a capsule, whereas in the other studies it was administered as a tablet. Each capsule or tablet contained 188.2 mg tafenoquine succinate equivalent to 150 mg of tafenoquine base. The methods used for the measurement of plasma tafenoquine concentrations have been published previously (*Thakkar et al., 2018*).

## Causal model of tafenoquine radical cure

The overall structure of the analysis is guided by the hypothetical simplified causal model shown in *Figure 4*. Under this model, the relationship between plasma tafenoquine concentrations and its radical curative activity is mediated by the production of oxidative intermediates (i.e. active metabolites). These are probably produced largely within hepatocytes, but also have systemic activity. These metabolites drive not only the efficacy of the drug but also the main adverse effects, notably haemolysis and the conversion of intraerythrocytic haemoglobin to MetHb. Thus, as MetHb production and haemolysis both result from the production of active oxidative metabolites, they could be proxy markers of radical curative efficacy (*White et al., 2022*).

## Population pharmacokinetic modelling

Tafenoquine pharmacokinetics have been described in detail previously (*Thakkar et al., 2018*). Tafenoquine is a slowly eliminated 8-aminoquinoline antimalarial drug with a terminal elimination half-life of around 15 days. The metabolites have not been well characterised. Tafenoquine pharmacokinetics have been characterised previously by a two-compartment pharmacokinetic (PK) model (i.e. bi-exponential decay model) with allometric scaling of clearance and volume parameters (*Thakkar et al., 2018*). We inspected visually all individual drug level profiles and excluded 10 concentrations from 8 patients which were substantial outliers. A total of 15 malaria patients vomited within one hour of drug administration. The protocol for all three studies indicated that these patients should be re-dosed at the same dose. One patient was not re-dosed and had very low tafenoquine levels during follow-up. We removed this patient from all analyses as it is difficult to estimate the amount of drug ingested. For the remaining 14 patients who vomited and were re-dosed, time since dose was calculated as the time since the second dose (in some patients this was the next day). All available tafenoquine concentration measurements were transformed into their natural logarithms and characterised using nonlinear mixed-effects modelling in the software NONMEM v7.4 (Icon Development Solution, Ellicott City, MD). A total of 2.56% of observed tafenoquine concentrations were measured to be below the limit of quantification and subsequently replaced with a value equal to half of the limit of quantification (*Beal, 2001*). One-, two-, and three-compartment dispositional models were evaluated, as well as different absorption models (i.e. first-order absorption and a fixed number of transit absorption compartments). Inter-individual variability was implemented as an exponential function of all structural PK parameters. The first-order conditional estimation method with interactions (FOCE-I) and subroutine ADVAN5 TRANS1 were used throughout the study. Body weight was implemented as a fixed allometric function on clearance parameters (exponent of 0.75) and as a linear function on volume parameters (exponent of 1.0). Additional, biologically relevant, covariates (i.e. age, sex, vomiting, tafenoquine formulation and patient vs healthy volunteers) were evaluated on all PK parameters using a linearized stepwise addition (p<0.01) and backward elimination (p<0.001) covariate approach. Statistically significant covariates retained in the backward elimination step were evaluated further for clinical significance and retained only if the range of individual covariate values resulted in a greater than +/-10% effect compared with the population mean value.

Potential model mis-specification and systematic errors were evaluated by basic goodness-of-fit diagnostics. Model robustness and parameter confidence intervals were evaluated by a sampling-important-resampling (SIR) procedure (*Dosne et al., 2017*). Predictive performance of the final model was illustrated by prediction-corrected visual predictive checks (n=2,000) (*Bergstrand et al., 2011*). The 5th, 50th, and 95th percentiles of the observed concentrations were overlaid with the 95% confidence intervals of each simulated percentile to detect model bias. Under the population mixed-effects model we estimated for each healthy volunteer and each patient in the three efficacy studies: (i) the area under the plasma concentration time curve ($\text{AUC}_{[0,\infty]}$); (ii) the maximum peak concentration ($C_{max}$); and (iii) the terminal elimination half-life (t1/2). $\text{AUC}_{[0,\infty]}$ and $C_{max}$ are summary statistics of the individual's exposure to the parent compound; whereas $t_{1/2}$ is a summary statistic of the metabolism of the parent compound (tafenoquine is eliminated via biliary excretion with enterohepatic recirculation, but is not eliminated in the urine *Brueckner et al., 1998*). Details of the final model are given in Appendix 6.

For the terminal elimination half-life, multiple patients had estimated values greater than 30 days. On examination, the NONMEM model estimates of the terminal elimination half-life were not robust with respect to the model structure. In order to derive robust half-life estimates, we fit a robust (using a student-*t* likelihood, *Lange et al., 1989*) Bayesian hierarchical linear model to the observed log-concentrations post day 4 (terminal phase; random intercept and slope). Patients had a median of 4 values post day 4 (IQR: 3–4). The half-life is then defined as $-\log(2)/\beta_i$ where $\beta_i$ is the individual slope estimate for patient $i$. Values below the lower limit of quantification were treated as left-censored observations with censoring value equal to log(2 ng/ml). The model included an interaction term between time and the log patient weight (clearance was associated with weight); and a dose (mg/kg) dependent intercept term. We then calculated the weight-adjusted terminal elimination half-life by subtracting the effect of weight estimated under a linear model.

The majority of patients in the three efficacy trials had between 5 and 6 observed plasma tafenoquine samples above the lower limit of quantification (*Appendix 1—figures 15 and 16*). We note that for patients with few samples (18 patients had fewer than 4 samples) the PK summary statistics ($\text{AUC}_{[0,\infty]}$, $C_{max}$, $t_{1/2}$) will be shrunk towards the population mean values.

## Methaemoglobin concentrations

All patients in the phase 1 trial and the three efficacy trials had frequent blood MetHb measurements taken using non-invasive signal extraction pulse CO-oximeter handheld machines. For the three efficacy trials, we excluded measurements taken at or after recurrence of vivax malaria. Because erroneous readings from the handheld oximeters can occur, we inspected visually the longitudinal MetHb curves for each patient. This visual check showed that all patients from one study site in Thailand had highly implausible measurements, most likely as a result of transcribing the carboxyhaemoglobin and not the MetHb value. All patients from this site were excluded from the MetHb analysis (*n*=91), resulting in a total of 9632 MetHb measurements across 746 *P. vivax* infected patients and 72 healthy volunteers. Between the time of first dose (defined as the time of the first chloroquine or DHA-PQP dose for the patients who did not receive tafenoquine, and the time of the first tafenoquine dose for those who did) and day 20, there were a total of 5816 measurements, with a median (range) of 7 measurements per patient (1–12).

Following our previous analysis, which examined the relationship between MetHb concentrations and *P. vivax* recurrence after high-dose primaquine (*Chu et al., 2021*), we summarised MetHb production using the day 7 value (approximately the day of the peak value). In patients who did not have a measurement on day 7, we used linear interpolation to estimate a day 7 value.

## Radical curative efficacy
### Rationale for the primary endpoint

The primary endpoint for the main efficacy analyses was any *P. vivax* recurrence within 4 months. We chose the binary endpoint of any recurrence rather than the time to first recurrence because tafenoquine has both asexual stage antimalarial activity (*Fukuda et al., 2017*) and hypnozoiticidal activity. Thus, an association between the mg/kg tafenoquine dose and the time to recurrence could, in theory, result from longer periods of post-treatment prophylaxis (i.e. suppression of blood stage multiplication) with higher mg/kg doses. Plasma concentrations above a plausible minimum inhibitory

concentration for the asexual activity persist for up to three months (**Dow and Smith, 2017**). Therefore, if relapses are delayed by the post-treatment prophylactic levels, we would expect to detect them before a 4-month cut-off, and definitely before a 6-month cutoff (ignoring long latency relapse, which will, in general, not be observed with 6 months follow-up). The choice of a particular cut-off for follow-up is thus a trade-off between sensitivity and specificity. Longer follow-up will increase the sensitivity of the endpoint (include more relapses) but decrease specificity (include more reinfections). We chose a 4-month cut off for the primary endpoint for the following two reasons:

- A priori we think that the 4-month endpoint will have very high sensitivity. This is >95% based on the data from the INSPECTOR trial. The INSPECTOR trial showed that in returning Indonesian soldiers, in whom reinfections were not possible, over 95% of all relapses following treatment with DHA-PQP plus tafenoquine occurred within 4 months (**Baird et al., 2020**). Very large field trials in patients given no radical cure and with multiple ACT partner drugs confirm the generalisability of this finding (**Taylor et al., 2019a**; **Chu et al., 2018**). For example, in the study by **Chu et al., 2018**, 90% of relapses were estimated to have occurred by 9 weeks. Extending the follow-up to 6 months will have minimal impact on sensitivity but will reduce specificity (by an amount that is directly proportional to local transmission intensity).
- The 4-month endpoint has a much lower proportion of loss-to-follow-up (i.e. incompletely captured data) compared to the 6-month endpoint. After excluding patients whose last visit was before day 31 (a total of 27 patients who are non-informative due to chloroquine post-treatment prophylaxis: no recurrences are expected in the first month **Taylor et al., 2019a**), only 12 out of the 1073 remaining patients did not have a 4-month follow-up visit (1.1%), compared to 98 out of 1073 who did not have 6-month follow-up visit (9.1%).

The main limitation of the binary endpoint of 'any recurrence' relates to the slightly longer post-prophylactic periods in the tafenoquine treated patients: their at-risk period for reinfection is shorter. All patients were treated with chloroquine after which the first detected recurrence (either reinfection or relapse) occurs approximately 1.5 months later (**White, 2011**). High doses of tafenoquine could plausibly delay recurrence for another 2 weeks (**Dow and Smith, 2017**). However, the post-prophylactic effect of both drugs most likely delays rather than suppresses reinfection occurring a few weeks after treatment (**Watson et al., 2018**). Thus, bias resulting from a greater number of reinfections in the placebo versus tafenoquine treated groups is likely to be small.

## Efficacy models

We explored the association between the following variables and the odds of *P. vivax* recurrence (the 12 patients with incomplete follow-up were coded as 'successes', i.e. no recurrent infection):

1. The mg/kg dose of tafenoquine, based on the recorded patient body weight (patients who received primaquine in comparator arms are included with an additional indicator variable multiplied by the mg/kg dose of primaquine);
2. The $AUC_{[0,\infty)}$ of the plasma tafenoquine concentration (only patients who received tafenoquine), estimated from the population PK model;
3. The tafenoquine $C_{max}$ (only patients who received tafenoquine), estimated from the population PK model;
4. The tafenoquine terminal elimination half-life $t_{1/2}$ (only patients who received tafenoquine), estimated from a robust linear model fit to the terminal phase data (post day 4), and with adjustment for weight;
5. The day 7 MetHb in a model adjusted for the mg/kg dose of tafenoquine (only patients who received tafenoquine);

For all predictors we fitted Bayesian mixed effects logistic regression models (random intercepts for study and study site), with adjustment for day 0 parasitaemia (log transformed). Logistic regression allows inference of the odds-ratio for recurrence for the predictive variable of interest, assuming a linear relationship between the predictive variable and log-odds for recurrence.

For the mg/kg dose (the primary predictor of interest), the dose-response relationship was then assessed under a Bayesian Emax model, where the dose-dependent logit probability of recurrence is defined as:

$$E_{max}(d, \mathbf{x}) = \alpha + \beta \mathbf{x} + \frac{\delta d^k}{d^k + \gamma^k},$$

where $d \geq 0$ is the tafenoquine dose (mg/kg) and $\mathbf{x}$ is a vector of additional covariates (for example, in the analyses which include data from primaquine treated patients, $\mathbf{x}$ would include the primaquine mg/kg dose); $\alpha$ is the intercept on the logit scale; $\delta$ determines the maximal tafenoquine effect (on the logit scale), i.e. the minimal logit probability of recurrence (an asymptote for very large doses); $k$ is the slope coefficient; $\gamma$ is the half-maximal effect (on the logit scale). The Emax model was also fitted with random intercepts for study and study site.

The Emax model is the most biologically plausible model for the pharmacodynamic effect of tafenoquine as estimated in these studies. No dose of tafenoquine is expected to reduce the number of recurrences by 4 months completely to zero as some of the recurrences will be reinfections.

### CYP2D6 polymorphisms

*CYP2D6* genotyping data were available for patients enrolled in DETECTIVE part 2 and GATHER. We mapped *CYP2D6* diplotypes to an activity score using the standard scoring system (*Caudle et al., 2020*) (note that under the updated scoring system the *10 allele has a score of 0.25 instead of 0.5). The activity score cannot be used as a linear predictor (e.g. a score of 1 cannot be interpreted as twice the metabolic activity of a score of 0.5). For this reason, to assess association with recurrence, we dichotomised the activity scores into poor metabolisers (≤0.5) versus normal/extensive metabolisers (>0.5) (*Caudle et al., 2020*).

### Comparison of predictors of recurrence

To determine the main predictors of recurrence, we fitted a multivariable penalised Bayesian logistic regression model. This included the following variables as predictors: the tafenoquine mg/kg dose; the plasma tafenoquine $AUC_{[0,\infty)}$; the plasma tafenoquine $C_{max}$; the day 7 MetHb concentration (%); the terminal elimination half-life $t_{1/2}$; and the baseline parasite density. The model included random intercept terms for each study and study site. Each fixed effect predictor was scaled to have mean 0 and standard deviation 1. To avoid convergence issues resulting from the co-linearity of the predictors, the prior on all non-hierarchical coefficients was set to a Normal(0,0.5). This prior effectively shrinks the regression coefficients towards zero. It can be interpreted as follows: plausible odds-ratios for recurrence for one standard deviation change for each of the fixed effect predictors lie approximately between 0.38 and 2.7).

### Sensitivity analyses

In a priori sensitivity analyses, all recurrences up until 6 months were included as both binary and time to event outcomes. The time to first recurrence was defined as a *P. vivax* episode between day 7 and day 180 with right censoring at the last recorded visit. We also re-ran the same models in patients who received a 300 mg single dose of tafenoquine only (i.e. the current recommended treatment).

### Tolerability and safety analyses

For tolerability, acute vomiting was defined as vomiting within 1 hr of tafenoquine dosing, and for safety, three definitions of severe haemolysis were assessed:

- Relative fall in haemoglobin by more than 25% to <7 g/dL
- Absolute fall in haemoglobin greater than 5 g/dL
- Any haemoglobin fall to <5 g/dL

Tolerability and safety events were categorised according to tafenoquine treatment and doses (no tafenoquine, <3.75 mg/kg tafenoquine, [3.75,6.25) mg/kg tafenoquine, [6.25,8.75) mg/kg tafenoquine, ≥8.75 mg/kg tafenoquine). The change in haemoglobin concentration from day 0 to the minimum on day 2 or 3, the expected day of nadir (*Commons et al., 2019*), was calculated to assess changes in haemoglobin associated with tafenoquine dose. Mixed effects models were used to explore the association between tafenoquine mg/kg dose, day 7 MetHb concentration, $AUC_{[0,\infty)}$, $C_{max}$, and $t_{1/2}$ adjusting for age, sex, day 0 parasite density and day 0 haemoglobin, with random effects for study and site.

### Statistical analyses

All analyses were done using R version 4.0.2 and Stata v17.0. Bayesian mixed-effects logistic regression models were fitted using the *rstanarm* package using default prior distributions. The Bayesian

Emax models were coded in stan and fitted using the *rstan* interface. Survival models were fitted in Stata. Population pharmacokinetic models were fitted using NONMEM v7.4. Terminal elimination half-lives were estimated using the *brms* package based on stan (*Bürkner, 2017*).

## Acknowledgements

We thank GlaxoSmithKline Research & Development Ltd for providing access to the clinical trial data via the https://www.clinicalstudydatarequest.com/ website. This work was supported by the Wellcome Trust. A CC BY or equivalent licence is applied to the author accepted manuscript arising from this submission, in accordance with the grant's open access conditions. NJW is a Principal Research Fellow funded by the Wellcome Trust (093956/Z/10 /C). JAW is a Sir Henry Dale Fellow funded by the Wellcome Trust (223253/Z/21/Z). RJC is funded by an Australian National Health and Medical Research (NHMRC) Emerging Leader Investigator Grant (1194702). RNP is a Wellcome Trust Senior Fellow in Clinical Science (200909). JAS is funded by an Australian NHMRC Leadership Investigator Grant (1196068).

## Additional information

### Competing interests

Robert J Commons: has received an Emerging Leader Investigator Grant (1194702) from the Australian National Health & Medical Research Council. The author has no other competing interests to declare. Joel Tarning: Joel Tarning has participated on the Novartis Malaria Advisory Council (payments made to their institution) and acts as an ASCPT Infectious Diseases steering committee member. The author has no other competing interests to declare. Justin A Green, Gavin CKW Koh: was formerly an employee of GlaxoSmithKline who funded the pre-registration studies of tafenoquine. The author holds stock in GlaxoSmithKline. The author has no other competing interests to declare. The other authors declare that no competing interests exist.

### Funding

| Funder | Grant reference number | Author |
| --- | --- | --- |
| Wellcome Trust | 223253/Z/21/Z | James A Watson |
| Wellcome Trust | 093956/Z/10/C | Nicholas J White |
| Australian NHMRC | 1194702 | Robert J Commons |
| Wellcome Trust | 200909 | Richard N Price |
| Australian NHMRC | 1196068 | Julie A Simpson |

The funders had no role in study design, data collection and interpretation, or the decision to submit the work for publication. For the purpose of Open Access, the authors have applied a CC BY public copyright license to any Author Accepted Manuscript version arising from this submission.

### Author contributions

James A Watson, Conceptualization, Software, Formal analysis, Investigation, Methodology, Writing – original draft, Project administration, Writing – review and editing; Robert J Commons, Resources, Data curation, Formal analysis, Methodology, Writing – review and editing; Joel Tarning, Formal analysis, Methodology, Writing – review and editing; Julie A Simpson, Formal analysis, Supervision, Investigation, Methodology, Writing – review and editing; Alejandro Llanos Cuentas, Resources, Validation, Writing – review and editing; Marcus VG Lacerda, Justin A Green, Gavin CKW Koh, Resources, Data curation, Writing – review and editing; Cindy S Chu, François H Nosten, Resources, Writing – review and editing; Richard N Price, Conceptualization, Supervision, Methodology, Writing – review and editing; Nicholas PJ Day, Conceptualization, Supervision, Validation, Investigation, Writing – review and editing; Nicholas J White, Conceptualization, Supervision, Validation, Investigation, Writing – original draft, Project administration, Writing – review and editing

## Author ORCIDs
James A Watson http://orcid.org/0000-0001-5524-0325
Robert J Commons http://orcid.org/0000-0002-3359-5632
Joel Tarning http://orcid.org/0000-0003-4566-4030
Julie A Simpson http://orcid.org/0000-0002-2660-2013
François H Nosten http://orcid.org/0000-0002-7951-0745
Richard N Price http://orcid.org/0000-0003-2000-2874
Nicholas PJ Day http://orcid.org/0000-0003-2309-1171
Nicholas J White http://orcid.org/0000-0002-1897-1978

## Ethics
Human subjects: Anonymised individual patient data were obtained via ClinicalStudyDataRequest.com following approval of a research proposal from the Independent Review Panel. Re-use of existing, appropriately anonymised, human data does not require ethical approval under the Oxford Tropical Research Ethics Committee regulations (OxTREC).

## Decision letter and Author response
Decision letter https://doi.org/10.7554/eLife.83433.sa1
Author response https://doi.org/10.7554/eLife.83433.sa2

---

# Additional files

## Supplementary files
• MDAR checklist

## Data availability
The data used in this study can be accessed by submitting a research proposal via https://www.clinicalstudydatarequest.com/ using the following study codes: GSK-TAF112582; GSK-200951; GSK-TAF116564. Decisions to share data with independent researchers are made via an Independent Review Panel. All code used to process the data is available on https://github.com/jwatowatson/Tafenoquine-efficacy, (copy archived at swh:1:rev:76f61918f2beaaaab0de4f5ccb3cf6cfc6503698).

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

## Appendix 1

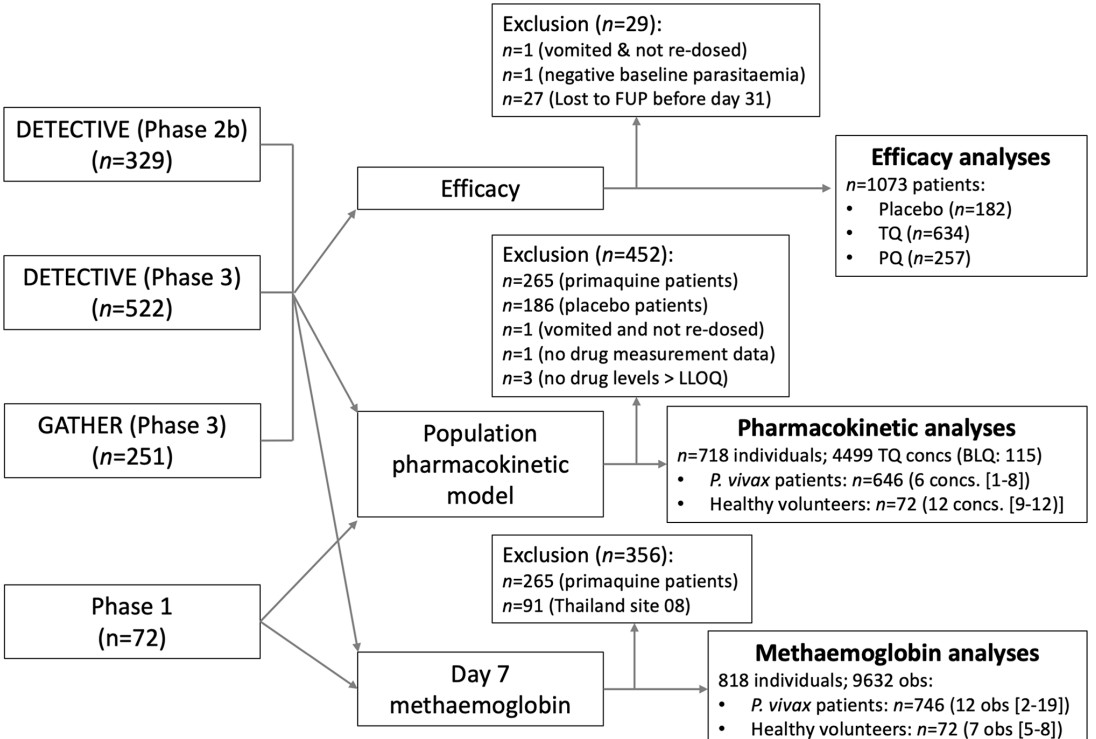

**Appendix 1—figure 1.** Flow diagram showing the construction of the analysis datasets for the binary endpoint efficacy analyses (note that the patients who dropped out before day 31 are included in the time to event analyses), the pharmacometric analyses $AUC_{[0,\infty)}$, Cmax, and $t_{1/2}$) and the day 7 methaemoglobin analyses.

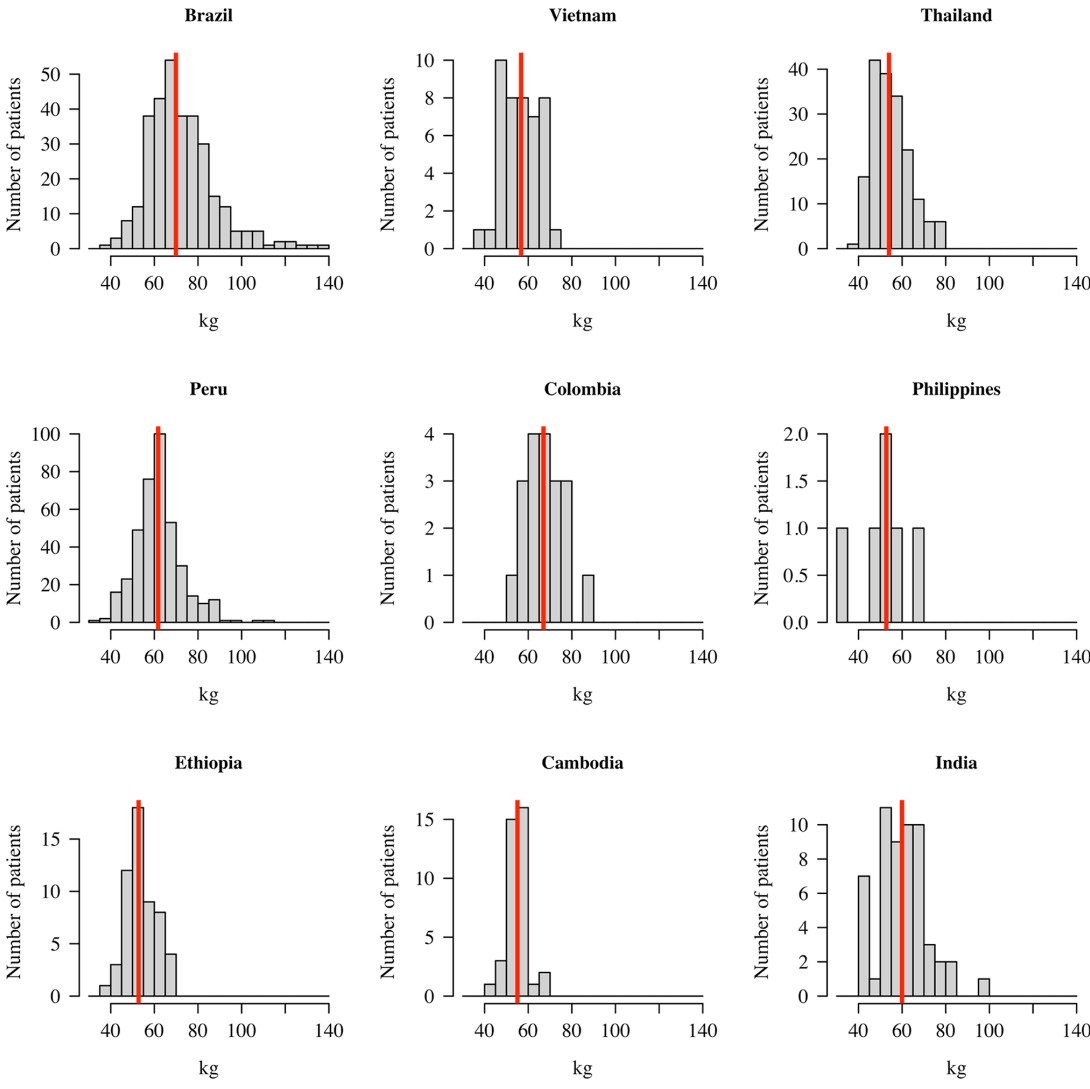

**Appendix 1—figure 2.** Distribution of patient weights across the nine countries. The vertical red lines show the median weights.

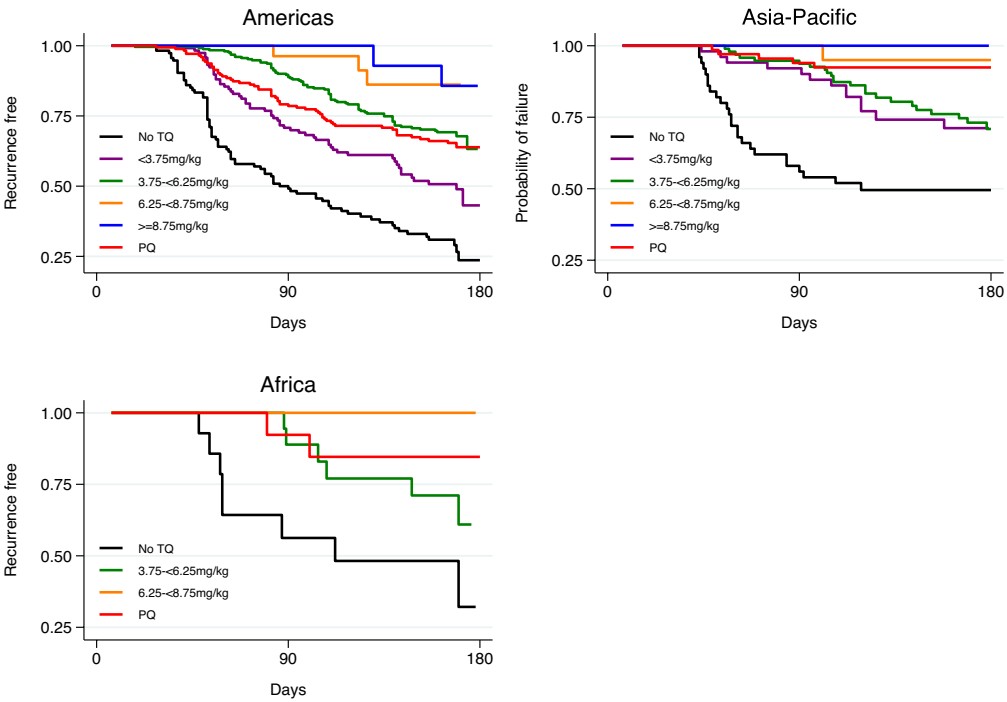

**Appendix 1—figure 3.** Kaplan-Meier survival curves for the time to first recurrence by region after administration of chloroquine without tafenoquine or with tafenoquine 300mg in *P.vivax* patients. Results include 455 patients from the Americas, 171 patients from Asia-Pacific and 42 patients from Africa. TQ: tafenoquine.

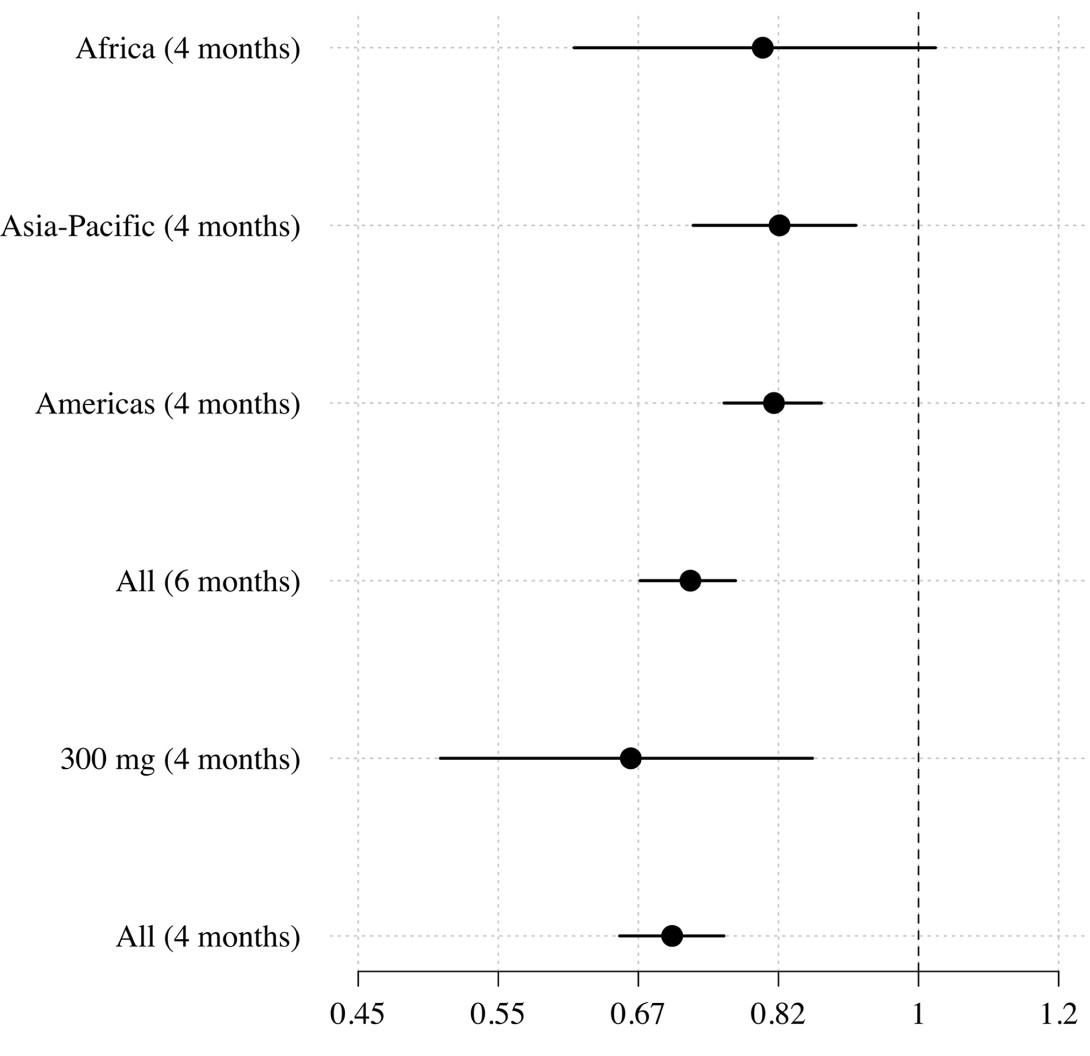

**Appendix 1—figure 4.** Comparison of estimated odds-ratios (95% CI) for the two endpoints (any recurrence by 4 months and any recurrence by 6 months); the sensitivity analysis in the 300mg group only; subgroups by geographic region.

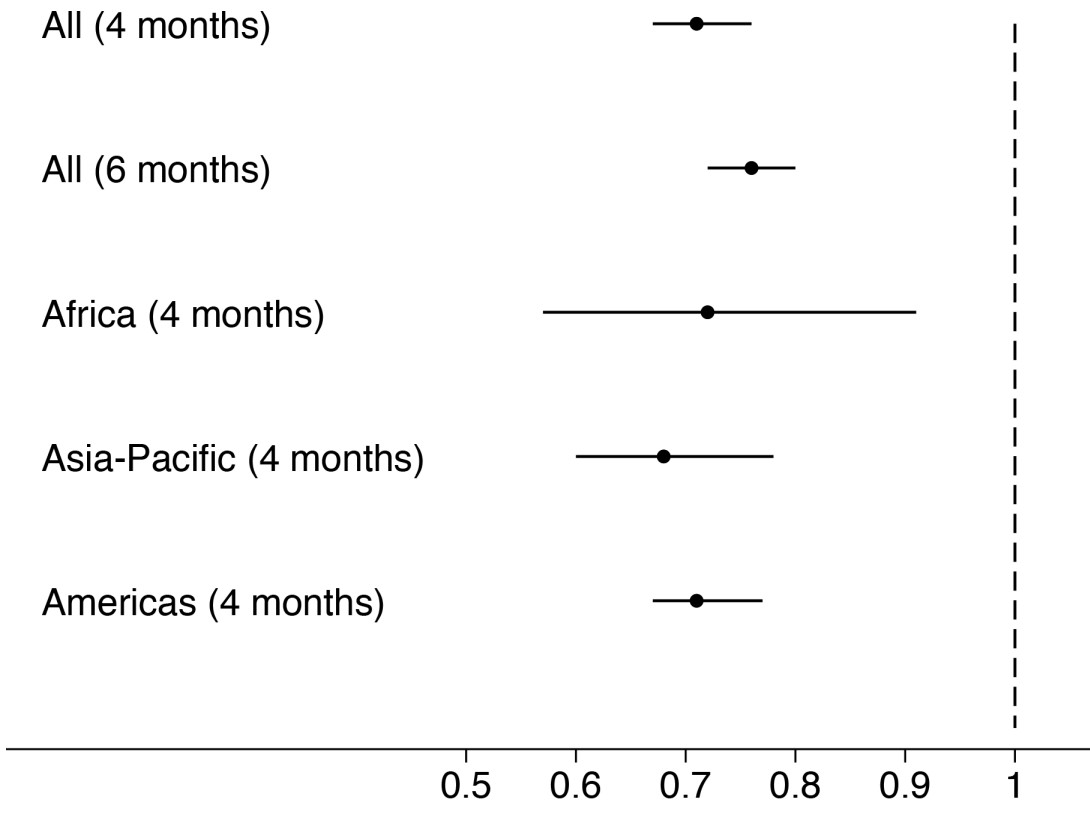

**Appendix 1—figure 5.** Comparison of estimated adjusted hazards ratio (95% CI) using time to event analysis for the two endpoints (any recurrence by 4 months and any recurrence by 6 months); subgroups by geographic region.

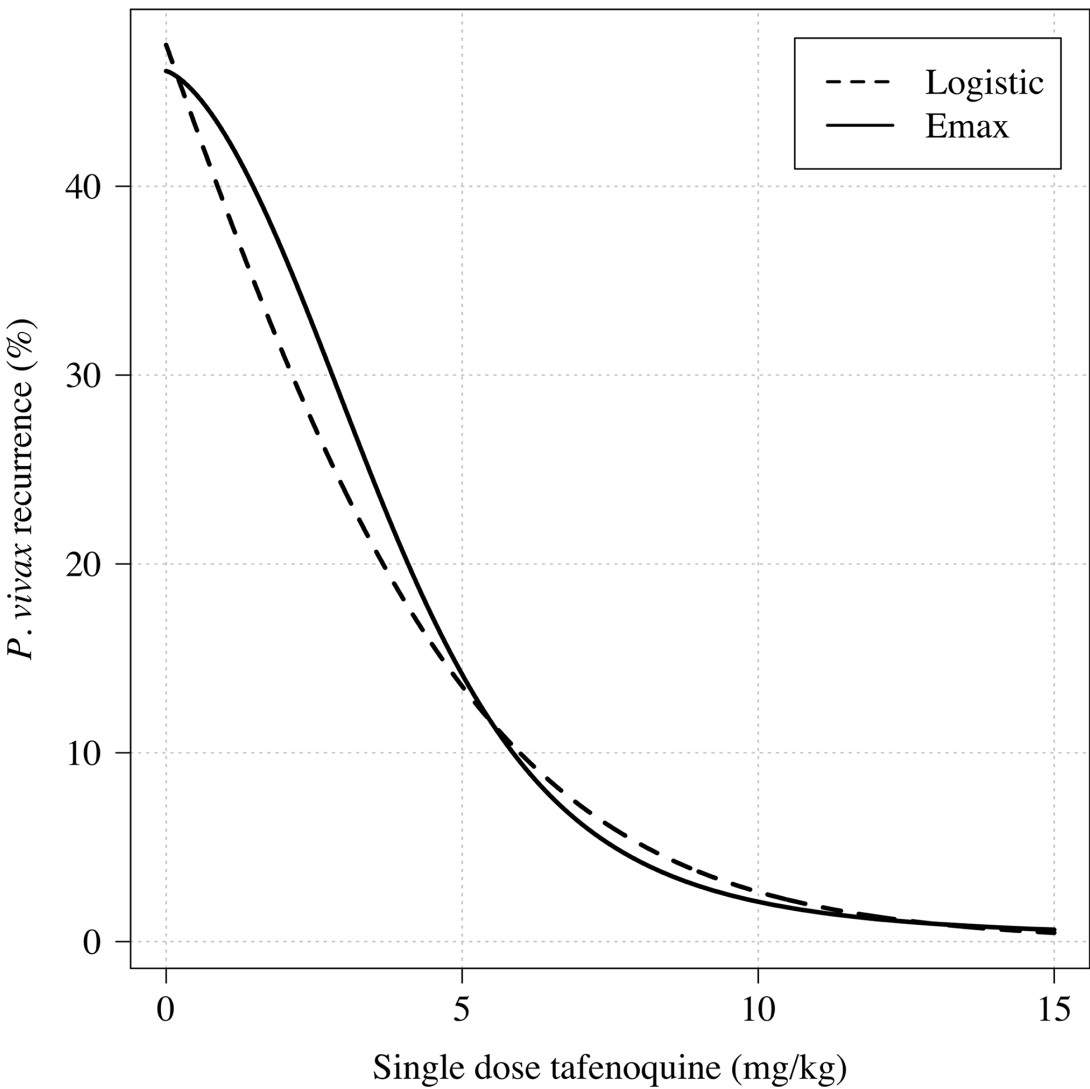

**Appendix 1—figure 6.** Comparison of logistic regression and Emax model fits for any recurrence within 4 months.

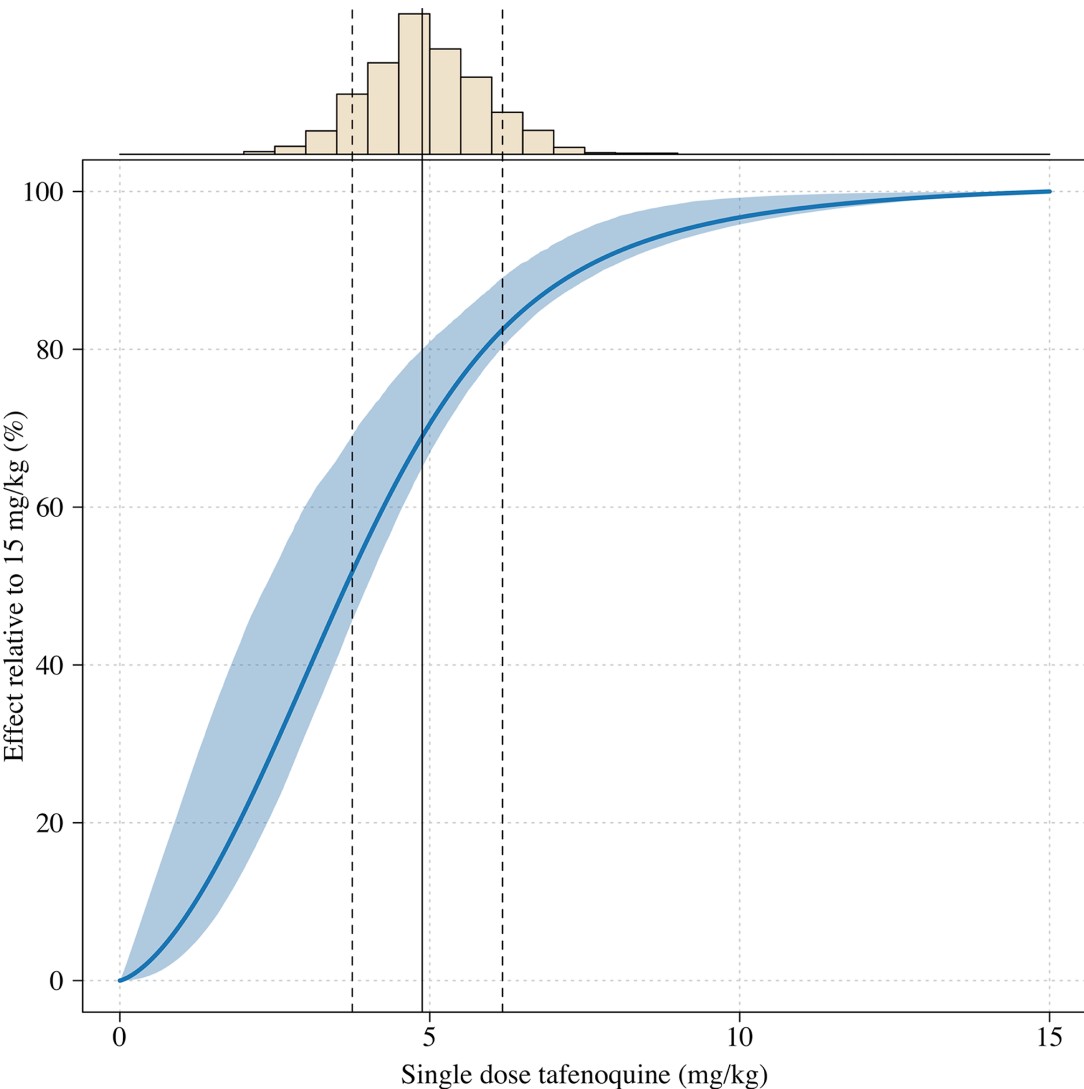

**Appendix 1—figure 7.** Estimated dose-response curve for tafenoquine mg/kg dose, whereby the maximal effect is defined as the effect for a 15mg/kg dose (the highest tolerated dose). The histogram above shows the distribution of mg/kg doses expected in patients using the empirical weight distribution from the three efficacy studies. The vertical lines show the 10, 50 and 90th percentiles of the expected mg/kg distribution.

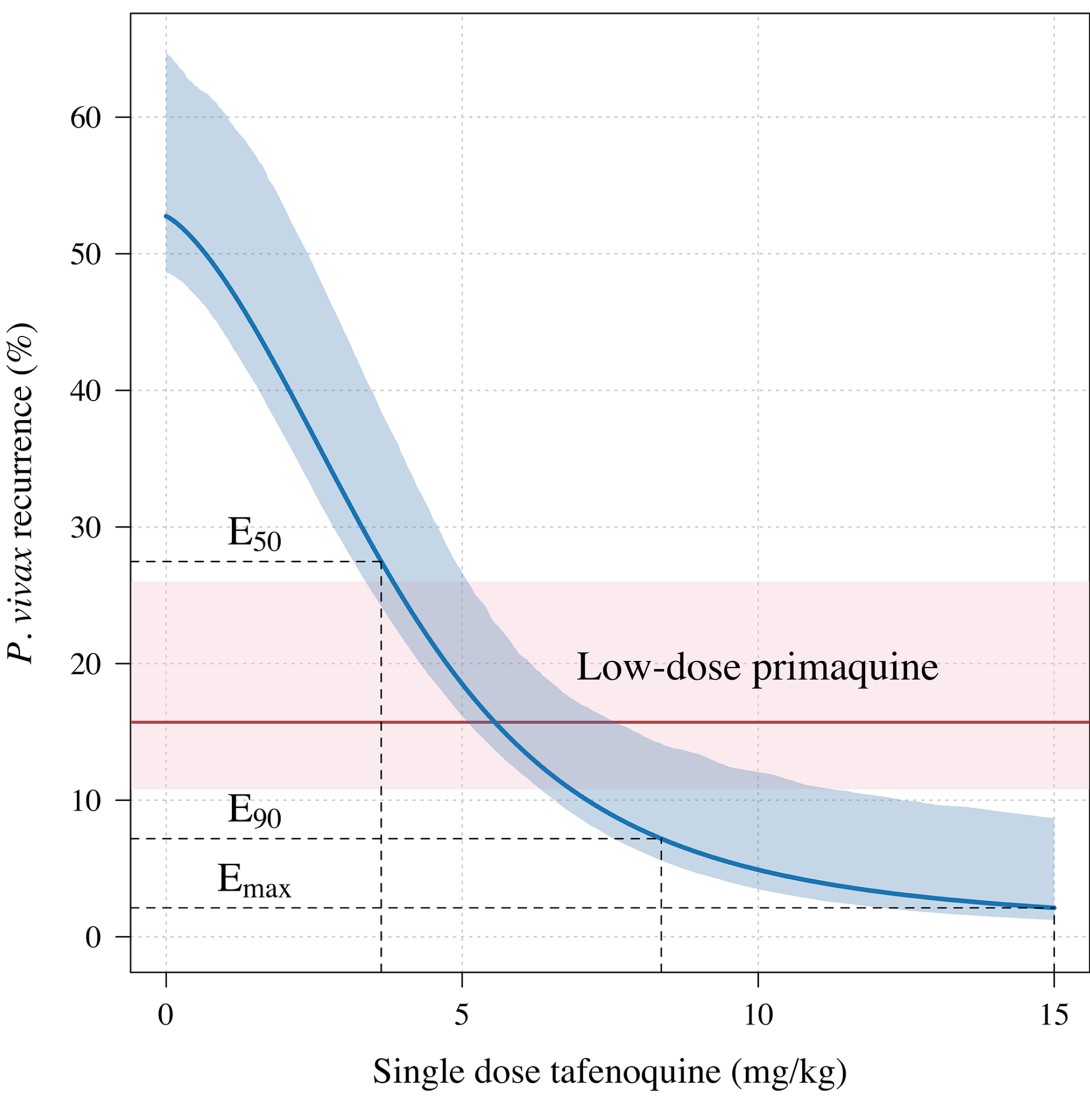

**Appendix 1—figure 8.** Tafenoquine mg/kg dose and the probability of any recurrence of *P.vivax* malaria at 6 months under the Emax model. Thick blue (shaded blue): Emax fit (95% CI) for any recurrence by 6 months; dashed red (shaded pink): estimated probability of any recurrence by 6 months after 3.5mg/kg primaquine (95% CI).

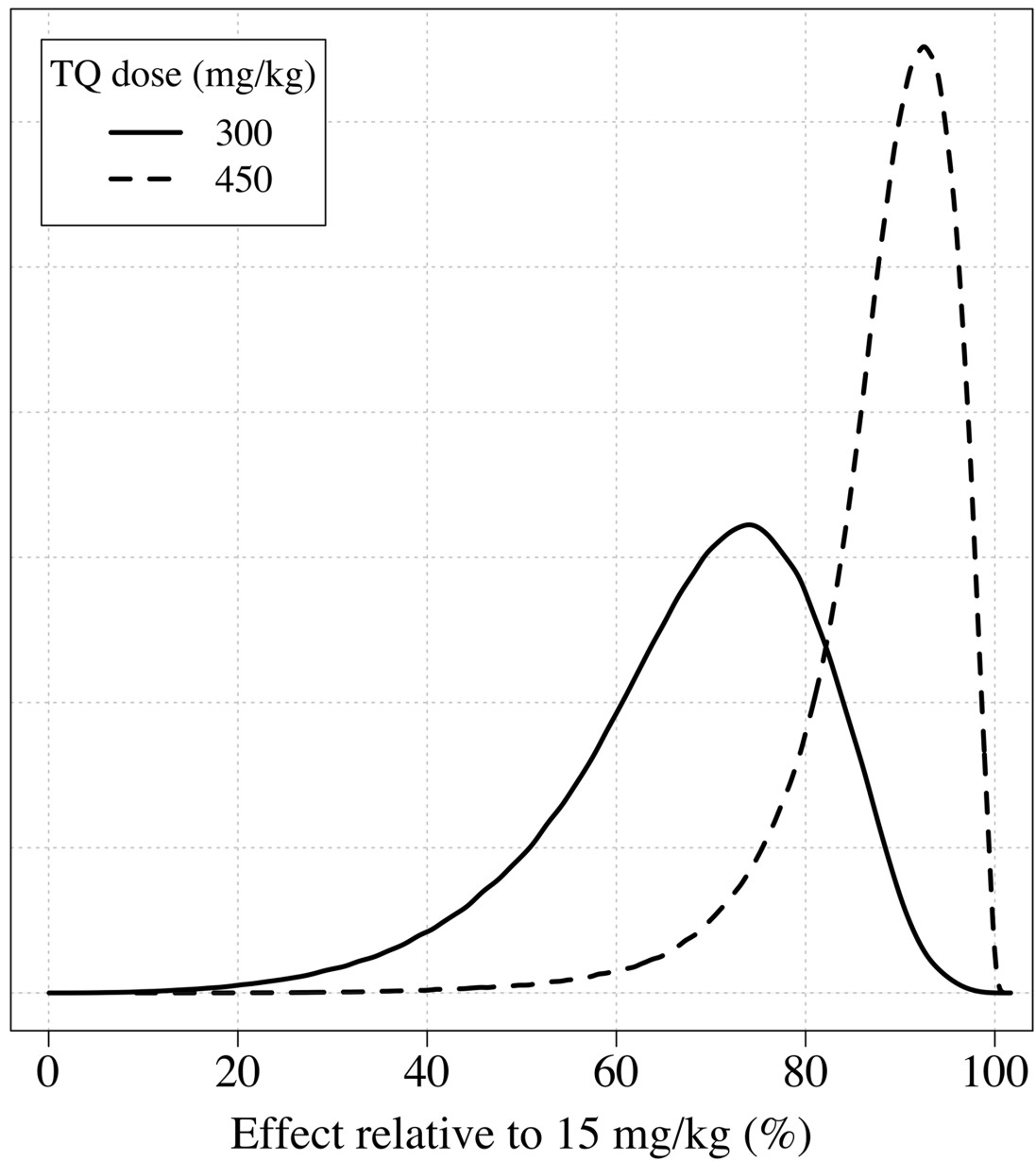

**Appendix 1—figure 9.** The distributions of the expected weight based efficacy (defined as the % reduction in recurrence at 4 months relative to the maximum dose of 15mg/kg) for a 300 mg single dose (thick line) and a 450mg fixed dose (dashed line). This is based on the empirical weight distribution from the three efficacy studies.

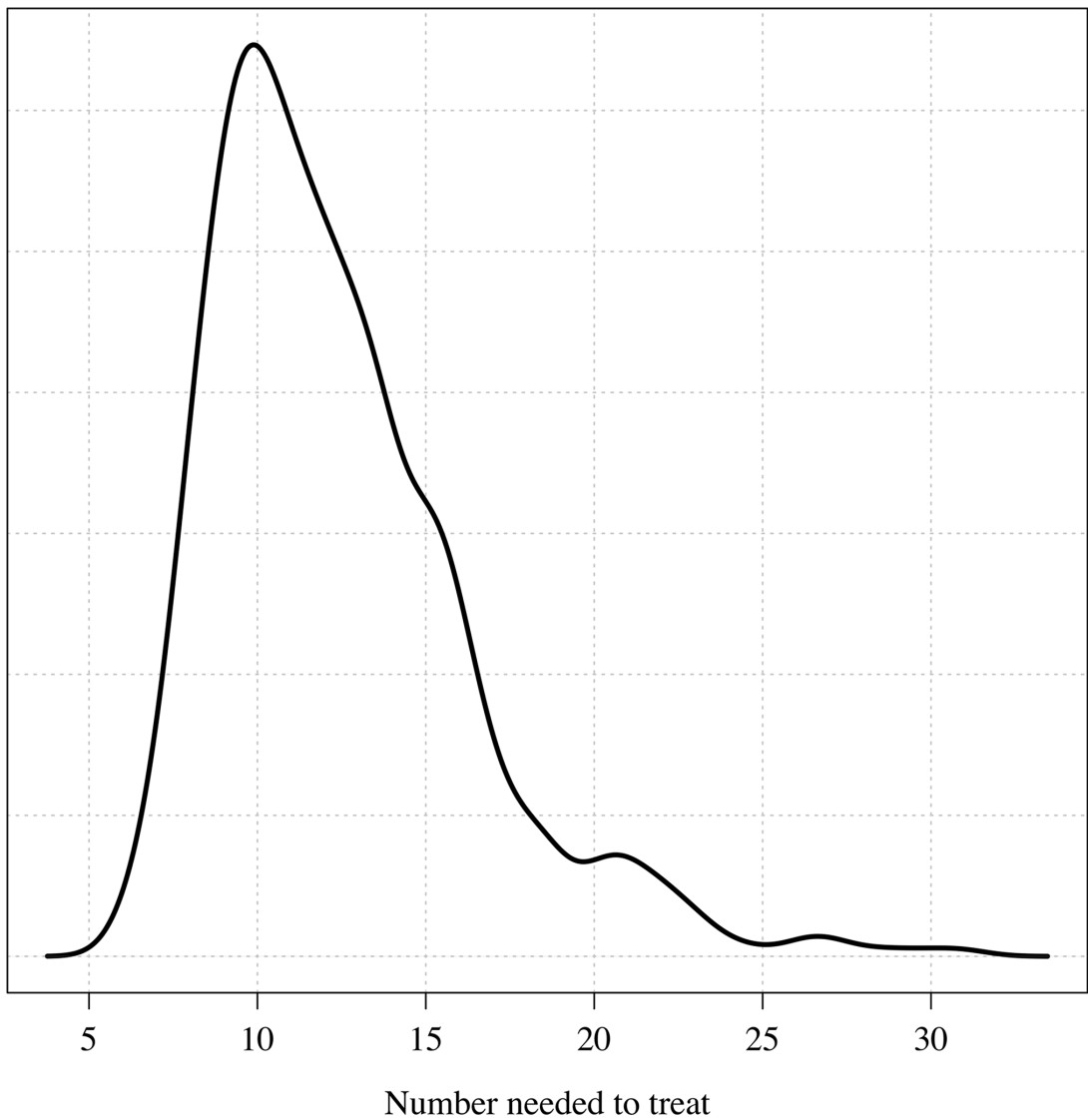

**Appendix 1—figure 10.** The posterior distribution over the number needed to treat to prevent one recurrence at 4 months under the Bayesian Emax model.

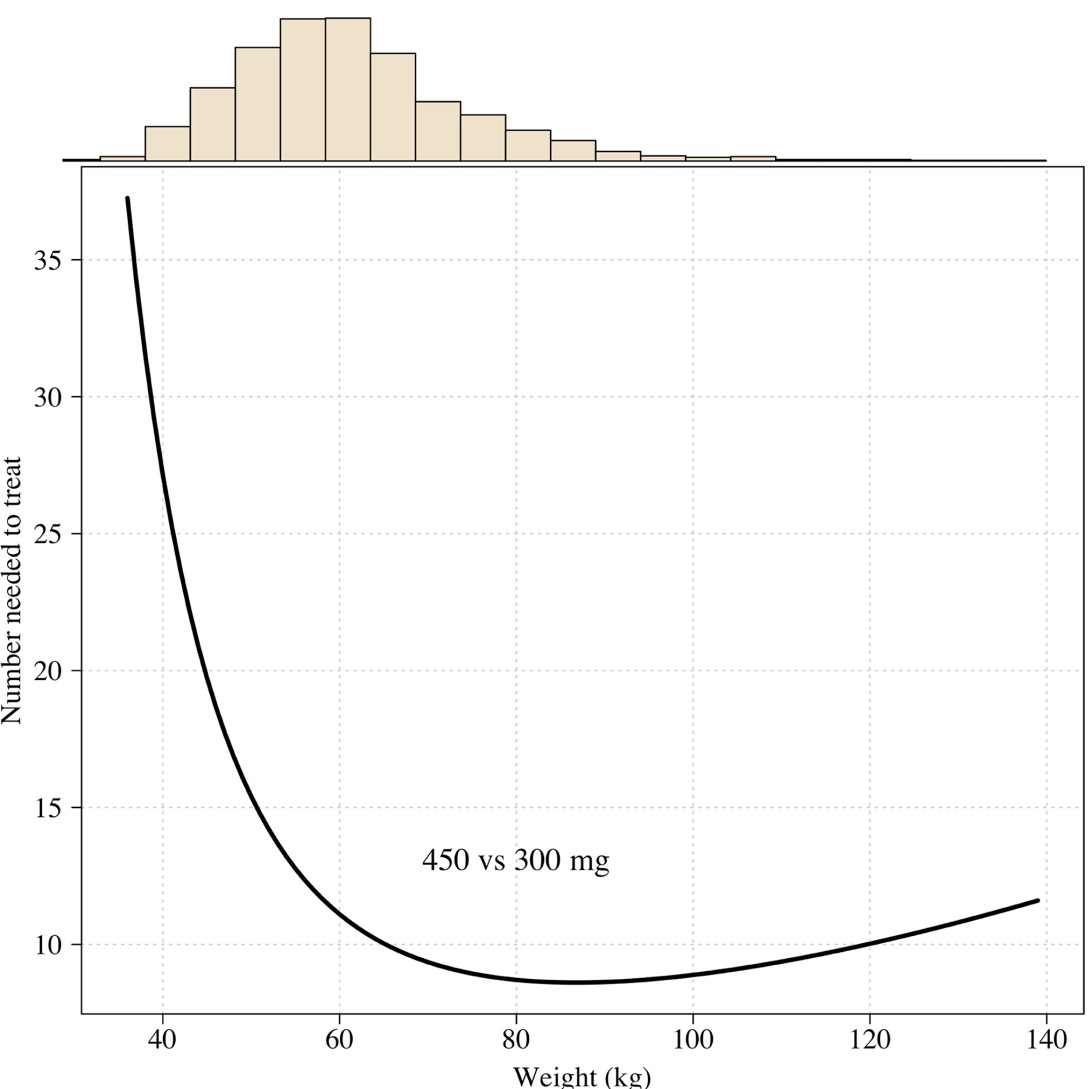

**Appendix 1—figure 11.** The number needed to treat with 450mg instead of 300mg to prevent one recurrence at 4 months as a function of patient weight under the Emax model.

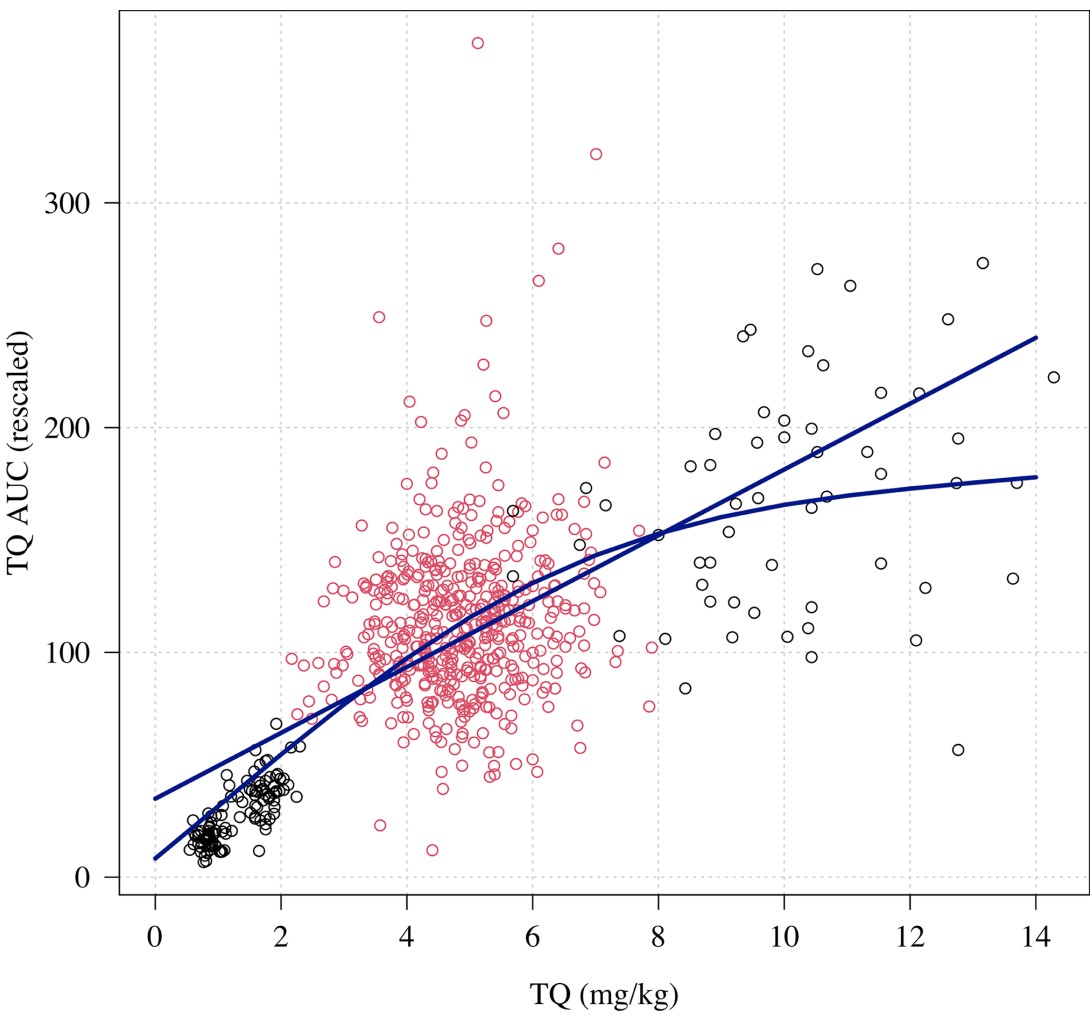

**Appendix 1—figure 12.** Relationship between mg/kg dose of tafenoquine and the $AUC_{[0,\infty)}$. Linear and spline fits are overlaid to show trend. The red dots show the patients who received a 300 mg dose (current recommended treatment), the black circles show the other doses (50, 100 and 600mg).

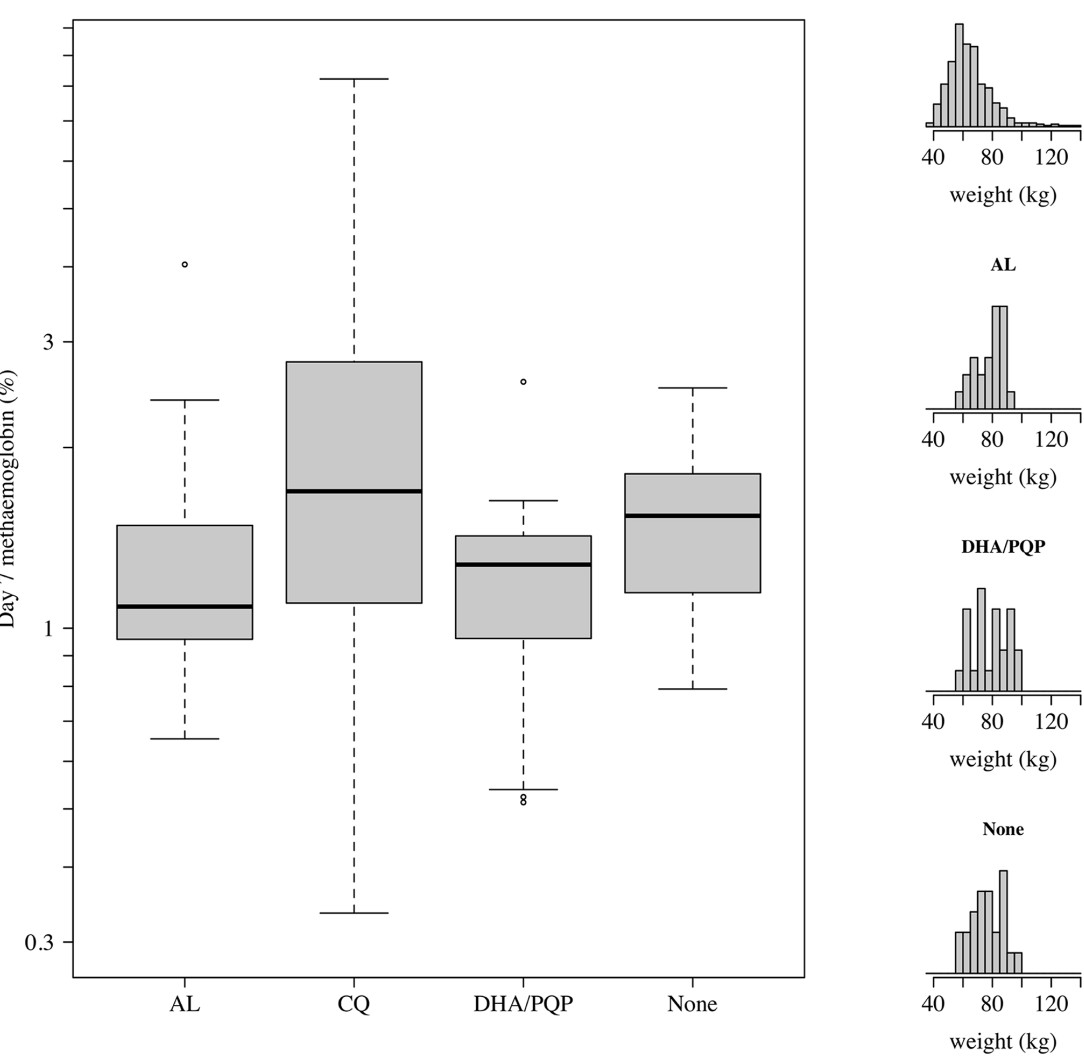

**Appendix 1—figure 13.** Comparison of day 7 methaemoglobin values after administration of tafenoquine 300 mg in *P.vivax* patients (all received chloroquine [CQ]) and healthy volunteers who were randomised to either no partner drug, AL, or DHA-PQP. The distributions of weights are shown in the right panels as the mg/kg dose is the primary driver of day 7 methaemoglobin. The mean weight in patients was slightly lower than in the healthy volunteers (although with a much larger variation).

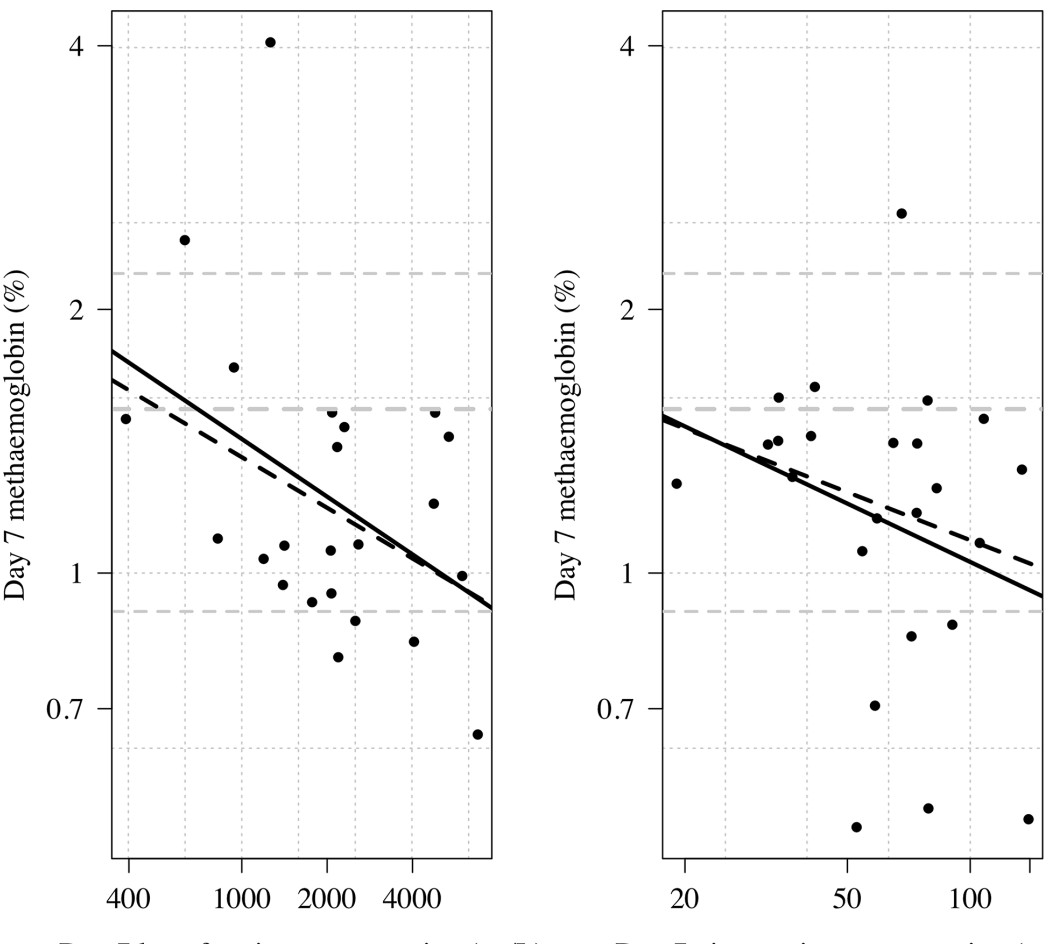

**Appendix 1—figure 14.** Day 7 methaemoglobin as a function of day 7 lumefantrine (n=23) or day 7 piperaquine concentrations (n=24) in the subjects randomised to tafenoquine +ACT in the phase 1 healthy volunteer trial (*Green et al., 2016*). The thick line shows a linear model fit; dashed line shows a robust linear model fit (rlm in MASS package in R). Both x and y axes are shown on the log scale. The grey lines show the median and 10/90th percentiles of the day 7 methaemoglobin levels in the tafenoquine only group (n=24).

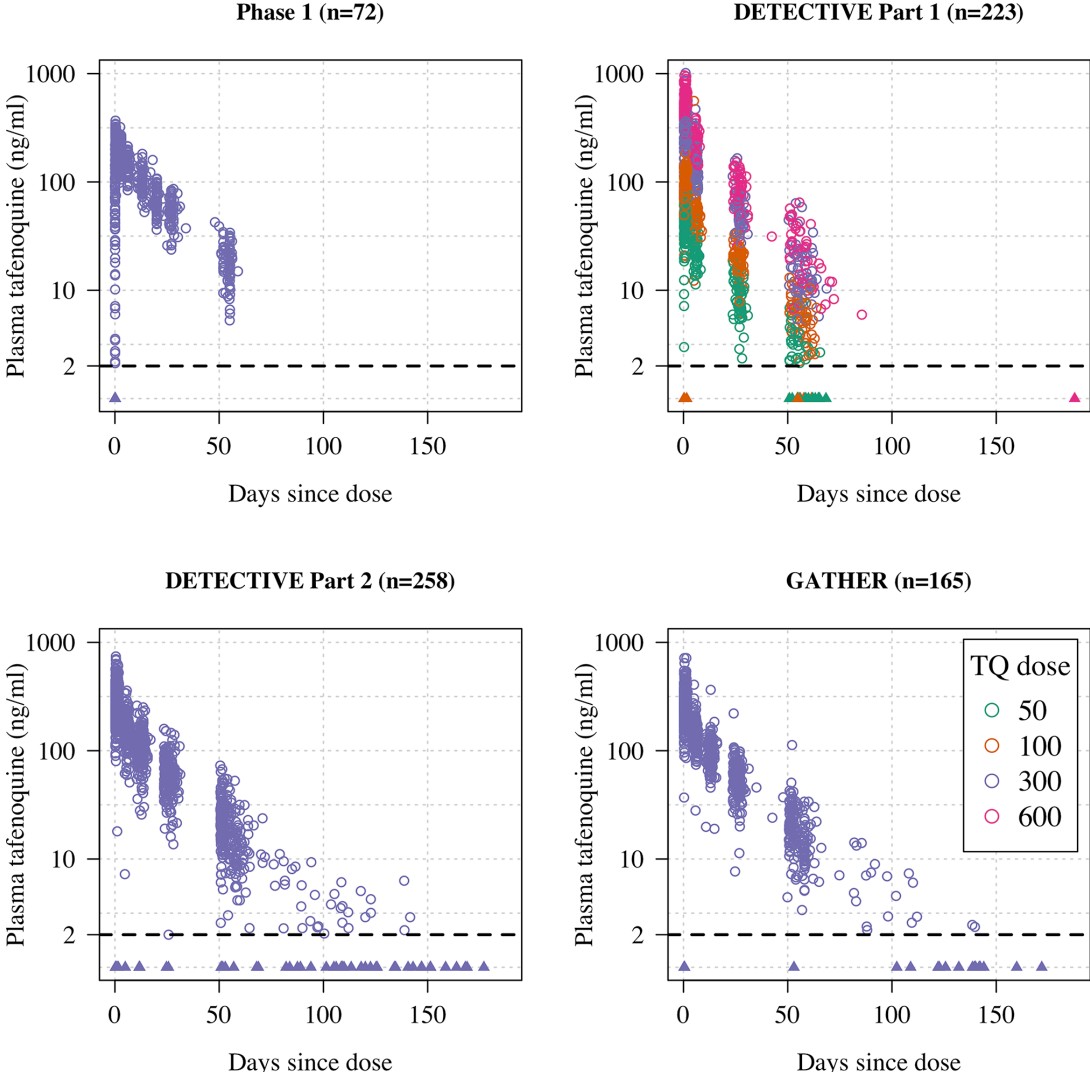

**Appendix 1—figure 15.** Plasma tafenoquine concentrations over time across the four studies (4499 concentrations in =*n*718 individuals). Concentrations below the lower level of quantification (BLQ) are shown by the triangles with a value of 1ng/ml (lower limit of quantification is shown by the dashed line at 2ng/ml). Circles are coloured by dose.

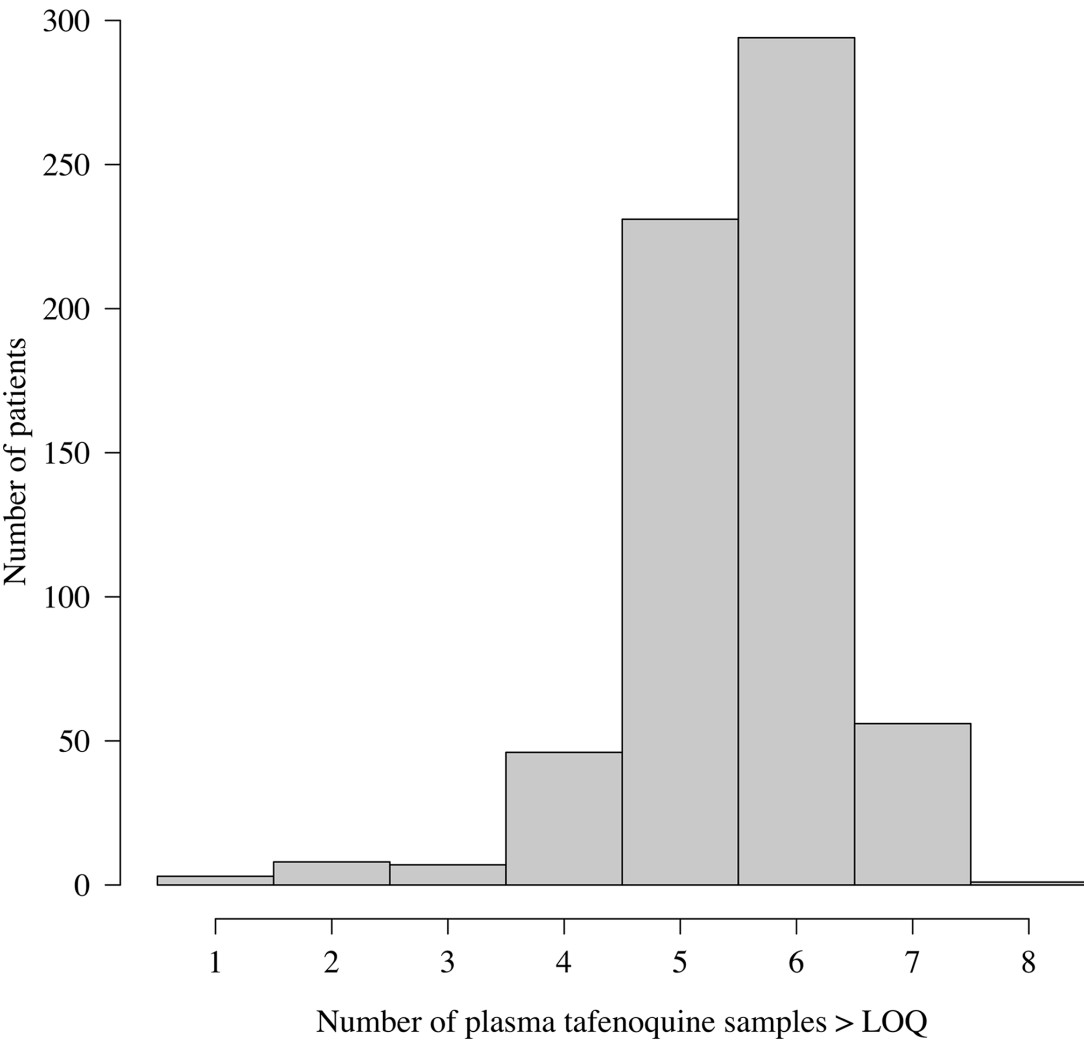

**Appendix 1—figure 16.** Distribution of the number of plasma tafenoquine concentrations per patient in the three efficacy trials. The very large majority of patients had 5 or 6 observed concentrations each. For those with few samples (e.g. 1–2), the estimates of the PK summary statistics ($AUC_{0,\infty}$, $C_{max}$ and $t_{1/2}$) will be shrunk towards the population mean estimates.

## Appendix 2

## Tafenoquine drug levels: data pre-processing

Phase 1 data

In the phase 1 study there were 860 drug level measurements in 72 patients. A total of 14 datapoints were below the lower limit of quantification (BLQ). The lower limit of quantification was 2 ng/ml. We removed one drug level which was a clear outlier from visual inspection, leading to a total of 859 levels in 72 patients in the analysis dataset.

Efficacy trials data

A total of 651 patients were randomised to tafenoquine in the three efficacy trials. One patient was excluded as they vomited but were not re-dosed. Out of the remaining 650 patient, drug measurement data were available from 649 patients. Three patients only had BLQ values and were removed from the analysis. 6 measurements were done in duplicate and the mean observed value was used (if one was BLQ then we imputed it as 1 ng/ml, i.e. half the lower limit of quantification). A total of 12 drug levels were removed after visual inspection of the individual profiles, leading to a total of 3640 measurements in 646 patients (101 BLQ).

## Appendix 3

### Comparison of major predictors of tafenoquine efficacy

We compared the predictive power of the tafenoquine mg/kg dose, the tafenoquine terminal elimination half-life, and the day 7 methaemoglobin concentration in a multivariable penalised logistic regression model (*n*=566 with complete data for all three predictors). Under this model, the mg/kg dose, the day 7 methaemoglobin and the terminal elimination half-life were all significantly associated with recurrence (*Appendix 3—figure 1*).

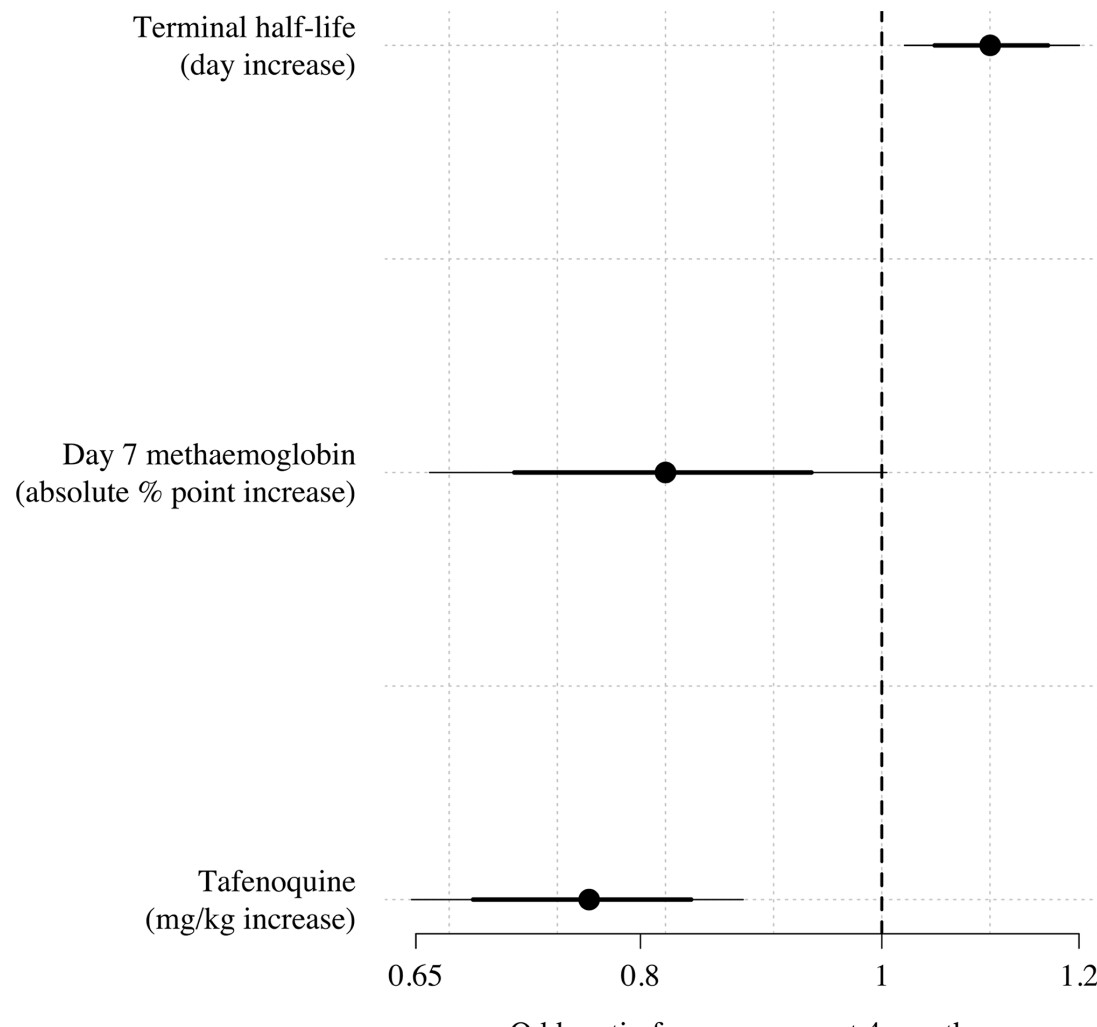

**Appendix 3—figure 1.** Multivariable model comparison of predictors of recurrence at 4 months in patients who received tafenoquine and who had recorded values (*n*=566). The circles (thick and thin lines) show the odds-ratios (80% and 95% credible intervals) for recurrence under a Bayesian penalised logistic regression model.

## Appendix 4

**Appendix 4—table 1.** Haematology safety outcomes by treatment arm.
n (%); mean (sd).

| Outcome | No TQ | <3.75 mg/kg | 3.75-<6.25 mg/kg | 6.25-<8.75 mg/kg | > 8.75 mg/kg |
|---|---|---|---|---|---|
| | n=186 | n=171 | n=378 | n=56 | n=44 |
| Hb fall >25% to<7 g/dL | 0 (0%) | 0 (0%) | 0 (0%) | 0 (0%) | 0 (0%) |
| Absolute Hb fall >5 g/dL | 0 (0%) | 0 (0%) | 1 (0.3%) | 0 (0%) | 0 (0%) |
| Hb fall to <5 g/dL | 0 (0%) | 0 (0%) | 0 (0%) | 0 (0%) | 0 (0%) |
| Hb change to day 2/3 | –0.4 (0.9) | –0.6 (0.9) | –0.7 (0.9) | –0.8 (0.8) | –0.6 (1.1) |
| Hb change to day 7±1 | –0.3 (0.9) | –0.3 (0.9) | –0.4 (1.1) | –0.4 (1.0) | –0.3 (1.3) |

## Appendix 5

### Systematic review of all published tafenoquine *P. vivax* treatment studies

The following four studies represent all published studies which administered tafenoquine as treatment (not including prophylaxis studies) which are not included in our individual patient data meta-analysis (in order of year of publication). They were extracted from the ongoing living systematic review of clinical efficacy studies of uncomplicated *P. vivax* by the WorldWide Antimalarial Resistance Network *Commons et al., 2017*.

1. *Walsh et al., 1999*: patients with *P. vivax* malaria were randomised to either (i) 300 mg daily for 7 days (n=15); (ii) 500 mg daily for 3 days (n=11); or (iii) 1 dose of 500 mg (n=9) (*Walsh et al., 1999*).
2. *Walsh et al., 2004*: patients with *P. vivax* malaria were randomised to either (i) 300 mg daily for 7 days (n=18); 600 mg daily for 3 days (n=19); 600 mg as a single dose (n=18) (*Walsh et al., 2004*).
3. *Fukuda et al., 2017*: patients with *P. vivax* malaria were randomised to either (i) 400 mg daily for 3 days (n=46) or (ii) chloroquine plus primaquine (*Fukuda et al., 2017*).
4. *Vélez et al., 2022* (TEACH trial): single arm paediatric cohort trial where tafenoquine dosing was based on weight. 14 patients received tafenoquine 100 mg; 5 received 150 mg; 22 received 200 mg; and 19 received 300 mg (*Vélez et al., 2022*).

The INSPECTOR trial has not been published, only presented at conferences (*Baird et al., 2020*).

Thus in total, 847 *P. vivax* malaria patients have received tafenoquine as treatment with total doses ranging from 100 to 2100 mg (daily doses ranging from 50 to 600 mg). Our data set thus represents 77% of all studied patients.

## Appendix 6

### Population pharmacokinetic analysis

Tafenoquine pharmacokinetics have been described previously by a two-compartment pharmacokinetic (PK) model (i.e. bi-exponential decay model) with first-order absorption and allometric scaling of clearance and volume parameters (*Thakkar et al., 2018*). This model was further improved by incorporating a transit-absorption model with 6 transit-compartments (ΔOFV: 1319). The absorption rate constant and the transit rate constant were assumed to be equal, thus resulting in no degree of freedom (df) difference to the traditionally used first-order absorption model. Adding an extra distribution compartment, resulting in a three-compartment distribution model, did not improve the model fit significantly (ΔOFV: 4.66; 2 df difference). The addition of body weight as an allometric function resulted in an improvement in model fit (ΔOFV: 80.1), and was also retained due to its biological relevance. The additional following covariate relationships were retained in the final model; *Age – CL/F*, impact of age on the oral elimination clearance implemented as a linear function, centered on 35 years of age (i.e. 4.34% decrease in CL/F per 10 years of age increase from the median age of 35 years). *Vomit – F*, impact of vomiting and redosing on relative oral bioavailability (i.e. 44.7% higher total dose when 1$^{st}$ dose was vomited and re-administered). *Patient – MTT*, impact of malaria on the mean absorption transit time (i.e. 93.7% faster drug absorption in patients compared to healthy volunteers). *Patient – Vc/F*, impact of malaria on the central volume of distribution (i.e. 27.6% lower central volume of distribution in patients compared to healthy volunteers). *Patient – Vp/F*, impact of malaria on the peripheral volume of distribution (i.e. 140% higher central volume of distribution in patients compared to healthy volunteers). Inter-individual variability in the central volume of distribution could be removed in the final model without a substantial impact on model fit (ΔOFV: 3.16). The final population pharmacokinetic model resulted in accurate and precise parameter estimates (*Appendix 6—table 1*) and adequate diagnostic performance (*Appendix 6—figure 1*) as well as predictive performance (*Appendix 6—figure 2*). Ignoring data below the lower limit of quantification did not impact the overall model performance, as the observed fraction of censored data was well described by the developed model (*Appendix 6—figure 3*).

**Appendix 6—table 1.** Final population pharmacokinetic parameter estimates of Tafenoquine.

| Pharmacokineticparameters | Population estimates *(%RSE [†]) | SIR median [†](95% CI [†]) |
|---|---|---|
| *Fixed effects* | | |
| F (%) | 1 *fixed* | |
| MTT (h) | 3.14 (12.8) | 3.05 (2.30–3.87) |
| CL/F (L/h) | 3.14 (8.43) | 3.14 (2.71–3.70) |
| Vc/F (L) | 1330 (12.9) | 1360 (1040–1710) |
| Q/F (L/h) | 5.82 (18.3) | 5.72 (3.99–8.05) |
| Vp/F (L) | 311 (14.4) | 318 (248–424) |
| *Covariate relationships* | | |
| Age – CL/F | –0.00434 (23.4) | –0.00435 (-0.00206–-0.00589) |
| Vomit – F | 0.447 (16.3) | 0.434 (0.315–0.608) |
| Patient – MTT | 0.937 (24.7) | 0.929 (0.605–1.51) |
| Patient – Vc/F | –0.276 (29.4) | –0.282 (-0.0784–-0.395) |
| Patient – Vp/F | 1.40 (21.6) | 1.30 (0.670–1.88) |
| *Random effects* | | |
| IIV F | 0.0850 (19.2) | 0.0870 (0.0559–0.120) |
| IIV MTT | 0.0831 (21.0) | 0.0887 (0.0585–0.124) |
| IIV CL/F | 0.0338 (18.8) | 0.0349 (0.0193–0.0451) |
| IIV Q/F | 0.151 (20.5) | 0.156 (0.1000–0.226) |

*Appendix 6—table 1 Continued on next page*

*Appendix 6—table 1 Continued*

| Pharmacokineticparameters | Population estimates *(%RSE [†]) | SIR median [†](95% CI [†]) |
|---|---|---|
| IIV Vp/F | 0.0256 (25.3) | 0.0265 (0.0153–0.0400) |
| RUV | 0.0225 (18.8) | 0.0240 (0.0170–0.0333) |

Abbreviations: SIR, sampling importance resampling; RSE, relative standard deviation; F, relative bioavailability; MTT, mean absorption transit time over 6 transit compartments; CL/F, oral elimination clearance; Vc/F, apparent central volume of distribution; Q/F, inter-compartment clearance; Vp/F, apparent peripheral volume of distribution; IIV, inter-individual variability presented as variance estimates; RUV, additive residual error presented as variance estimates.

Covariate relationships: *Age – CL/F*, impact of age on the oral elimination clearance implemented as a linear function, centered on 35 years of age (1 + Θ × (Age-35)). *Vomit – F*, impact of vomiting and redosing on relative oral bioavailability (Vomiting: 1 + Θ). *Patient – MTT*, impact of malaria on the mean absorption transit time (Malaria patients: 1 + Θ). *Patient – Vc/F*, impact of malaria on the central volume of distribution (Malaria patients: 1 + Θ). *Patient – Vp/F*, impact of malaria on the peripheral volume of distribution (Malaria patients: 1 + Θ).

*Computed population mean parameter estimates from NONMEM were calculated for a typical healthy individual with a body weight of 70 kg and 35 years of age.

[†]Computed from sampling importance resampling (SIR; 1,000 samples, 200 resamples) and presented as median estimate (2.5[th] to 97.5[th] percentiles).

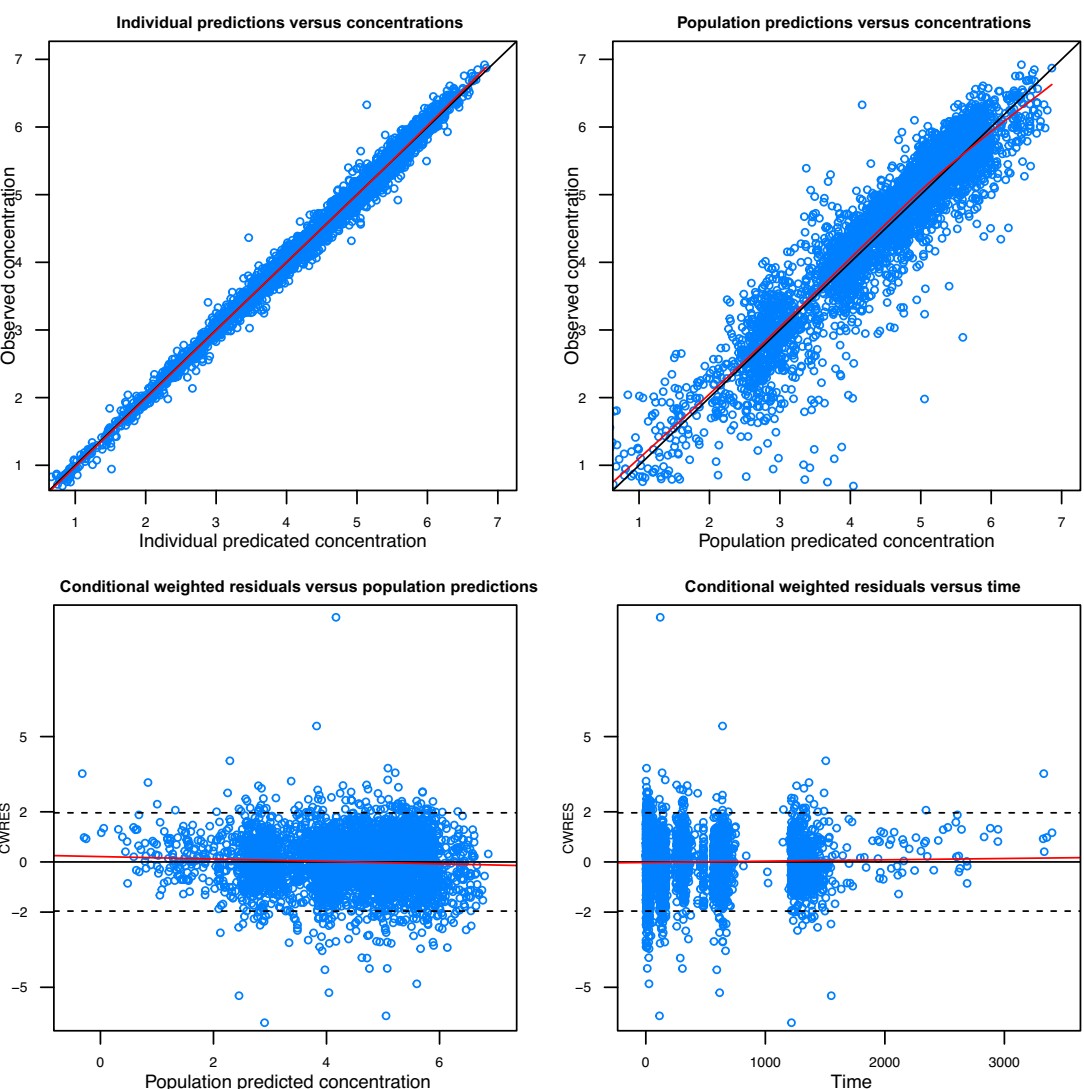

**Appendix 6—figure 1.** Goodness-of-fit diagnostics of the final population pharmacokinetic model of tafenoquine. Observations are represented by open blue circles, solid black lines represent the line of identity or zero line, and the local polynomial regression fitting for all observations are represented by the solid red lines.

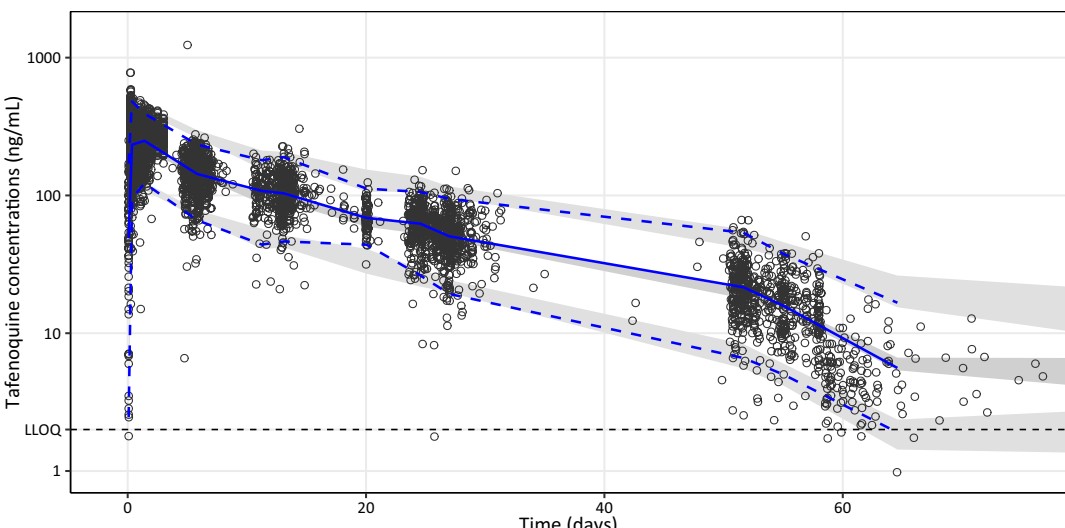

**Appendix 6—figure 2.** Visual predictive plot of the final population pharmacokinetic model of tafenoquine. Open black circles represent observed plasma concentrations. Solid and dashed blue lines represent the median, 5th, and 95th percentiles of the observations. The shaded grey areas represent the predictive 95% confidence intervals around the simulated 5th, 50th, and 95th percentiles (n = 2,000 simulations). The broken horizontal line represents the lower limit of quantification (LLOQ) of 2 ng/mL.

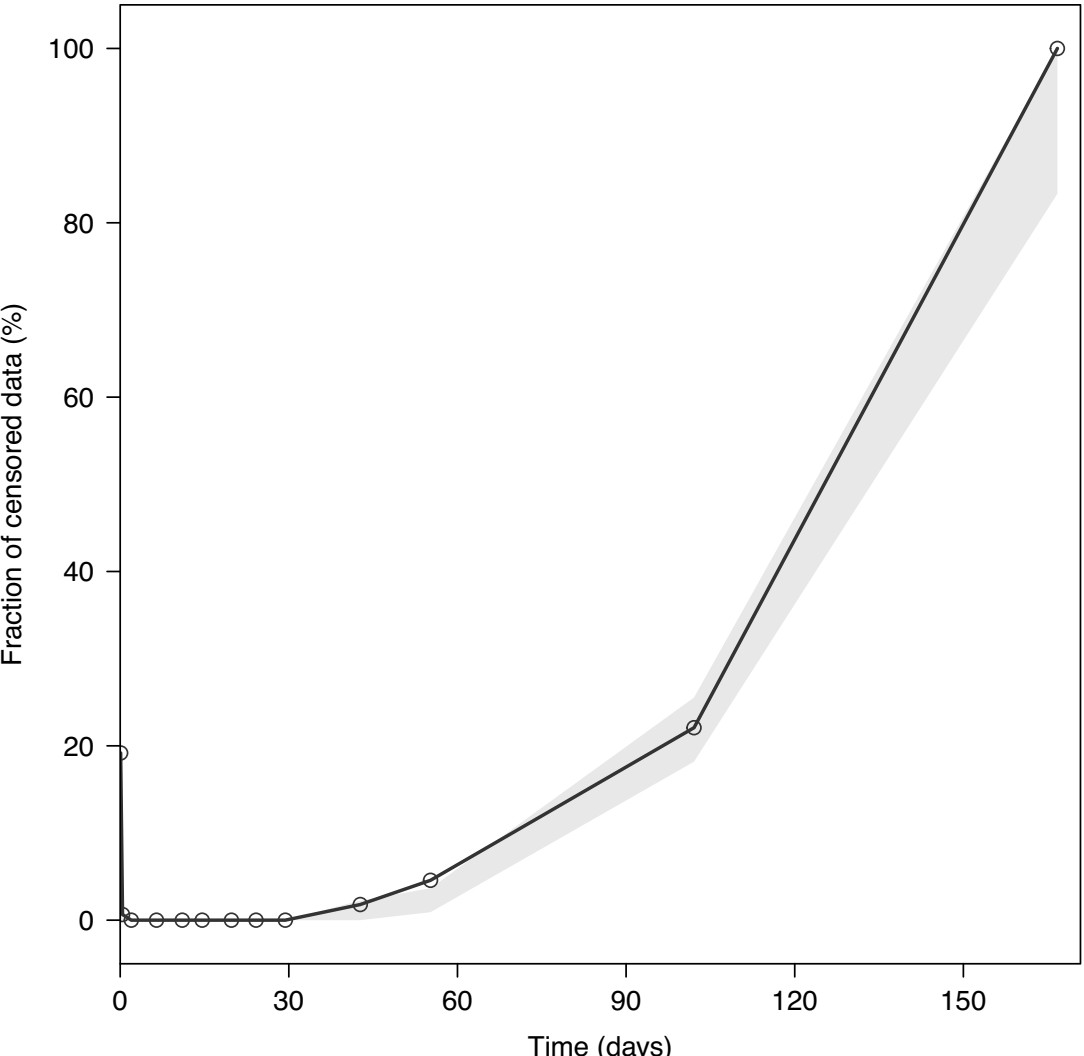

**Appendix 6—figure 3.** Visual predictive plot of the fraction of censored tafenoquine concentrations. The solid black line represents the observed fraction of censored data (i.e. data below the lower limit of quantification). The shaded grey area represents the predictive 95% confidence intervals of the simulated fraction of censored data (n = 2,000 simulations).

