## [Editor Report]

This competently performed retrospective analysis presents important findings concerning the clinical use of tafenoquine, a drug against Plasmodium vivax malaria. The assembly of the majority of global tafenoquine pharmacology data from clinical treatment studies provides compelling evidence in support of the drug's regimen that includes an increase in dosing, which would lead to a significant enhancement of the drug efficacy, hence a decrease in recurrent parasitemia. The manuscript includes a detailed analysis and discussion concerning the side effects of the drug affecting more susceptible populations.

---

## [Decision Letter]

**Decision letter after peer review:**

Thank you for submitting your article "The clinical pharmacology of tafenoquine in the radical cure of *Plasmodium vivax* malaria: an individual patient data meta-analysis" for consideration by *eLife*. Your article has been reviewed by 3 peer reviewers, and the evaluation has been overseen by a Reviewing Editor and Dominique Soldati-Favre as the Senior Editor. The following individuals involved in the review of your submission have agreed to reveal their identity: Jose Pedro Gil (Reviewer #1); Abdre Daher (Reviewer #2); G. Dennis Shanks (Reviewer #3).

Essential revisions:

1) Given the nature of your study and the detailed comments from the 3 Reviewers, I would like to suggest that you respond to the points raised in the reviews as you deem appropriate to improve the presentation of your retrospective meta-analysis.

2) Amongst other revisions, please include a more detailed analysis and discussion concerning the importance (or lack of it) of the patient CYP2D6 status in Tafenoquine T_1/2_, methemoglobin levels, and overall efficacy as well as some potential side effects, particularly in vulnerable populations.

*Reviewer #1 (Recommendations for the authors):*

I do not have any significant comments concerning the PK/PD modelling; they are clear and well-explored taking into consideration the retrospective nature of the report and the available data. That said, I feel that the manuscript can be improved, especially in what concerns the possibly overlooked importance of CYP2D6.

Some comments/suggestions:

"Under the model to all data (n=1,073), each additional mg/kg of tafenoquine was associated with an odds ratio (OR) for recurrence within 4 months of 0.70 (95% CI: 0.65 to 0.76). In comparison, each additional mg/kg of primaquine was associated with an odds ratio of 0.62 (95% CI 0.55 to 0.71) each absolute percentage point increase in the day."

It would be informative to show that the incremental performance of both drugs is not statistically different. At a first glance, it seems that primaquine gives better protection for each increase in mg/Kg. The operational advantage of tafenoquine is obvious (uni-dose), but the simple comparison between the two regimens suggests that primaquine has a better performance. If this is the reality, it should be clearer from the text, as well as a reference to the final cost-benefit balance, more favourable to Tafenoquine.

"MetHb concentration was associated with an odds ratio for recurrence of 0.81 (95% CI 0.65 to 0.99). Furthermore, consistent with this association between MetHb and vivax malaria recurrence (Figure 4), the day 7 MetHb levels were weakly inversely correlated with the tafenoquine weight-adjusted t1_2 values (Figure 3d; ⇢ = *0.1, p=0.05)."

Production of methemoglobin is associated with the therapeutic effects of tafenoquine (and primaquine). This might be related to the fact that the tafenoquine (and primaquine) clinically active metabolites that kill the parasite are also involved in oxidative stress creating higher levels of methemoglobin. In this scenario, I suggest that the authors investigate associations between the available CYP2D6 status and D7 methemoglobin levels. Metabolism affects the T_1/2_ of a drug, and knowing this PK factor is associated with increased efficacy, the hypothesis of discerning a CYP2D6 influence is in my view interesting. Further, this information should be incorporated into the presented PD modelling. In a nutshell, are different CYP2D6 allele statuses related to different curves of incremented efficacy (OR ~ δ Mg/Kg)? Are there human populations "promoting" a better tafenoquine performance, or is this associated with increased adverse events affecting the final cost-benefit of increasing the dosing?

In this context, I regret the lack of information concerning CYP2D6 copy number variation, a mutation relatively common in SE Asia. This is an obvious consequence of the limitations of retrospective studies.

Finally, it is to note that CYP2D6 activity scores are derived from experimentation using specific probe drugs. Albeit generally informative, these scores (a quite rough metric on their own) might not reflect the enzyme capacities when considering other substrates, such as the 8-aminoquinolines. Also, the data presented, especially when concerning CYP2D6 has to be considered partial. This gene is highly variable, and only a full ORF-phased sequencing can establish its true primary sequence nature (e.g. see Gaedick et al., 2018, J Pers Med. 2018 Apr 17;8(2):15. doi: 10.3390/jpm8020015)

"The phase 1 study was a drug-drug interaction study, with volunteers randomised to either tafenoquine alone or tafenoquine in combination with DHA-PQP or artemether-lumefantrine (AL), the standard antimalarial treatment regimens in both cases."

Lumefantrine is well known to be a competent CYP2D6 inhibitor. I did not find any indication in the report of this factor being explored and incorporated into the overall model. Can the authors elaborate on the data concerning AL-treated subgroups?

*Reviewer #2 (Recommendations for the authors):*

Congratulations on the paper.

The sensitive analysis for the 4 vs 6 months outcome could be further emphasized, as shortening the time to the last outcome can be very useful for future clinical trials.

It would be interesting to have a prediction of the haemolysis in case of the unfortunate inadvertent use of the new dose in G6PD deficient female heterozygotes. This would be the primary clinical concern in remote areas.

The conclusion states: Patient weight was not a predictor of recurrence per se. This could be clarified.

*Reviewer #3 (Recommendations for the authors):*

I recommend the article be published largely as it is written.

Authors may respond to some of the points raised in the public review as appropriate.

---

## [Author Response]

Essential revisions:1) Given the nature of your study and the detailed comments from the 3 Reviewers, I would like to suggest that you respond to the points raised in the reviews as you deem appropriate to improve the presentation of your retrospective meta-analysis.

We have added to the Discussion and the Results (exact additions given in the point-by-point replies).

2) Amongst other revisions, please include a more detailed analysis and discussion concerning the importance (or lack of it) of the patient CYP2D6 status in Tafenoquine T_1/2_, methemoglobin levels, and overall efficacy as well as some potential side effects, particularly in vulnerable populations.

We have added:

An analysis looking at *CYP2D6* activity score vs tafenoquine half-life and a paragraph in the Discussion (Limitations);

An analysis of day 7 methaemoglobin and exposure to lumefantrine and piperaquine in the phase 1 trial;

A review of all studies that have given tafenoquine as treatment for vivax malaria. This shows we have 77% of all studied patients in our individual patient data meta-analysis.

Reviewer #1 (Recommendations for the authors):I do not have any significant comments concerning the PK/PD modelling; they are clear and well-explored taking into consideration the retrospective nature of the report and the available data. That said, I feel that the manuscript can be improved, especially in what concerns the possibly overlooked importance of CYP2D6.Some comments/suggestions:"Under the model to all data (n=1,073), each additional mg/kg of tafenoquine was associated with an odds ratio (OR) for recurrence within 4 months of 0.70 (95% CI: 0.65 to 0.76). In comparison, each additional mg/kg of primaquine was associated with an odds ratio of 0.62 (95% CI 0.55 to 0.71) each absolute percentage point increase in the day."It would be informative to show that the incremental performance of both drugs is not statistically different. At a first glance, it seems that primaquine gives better protection for each increase in mg/Kg. The operational advantage of tafenoquine is obvious (uni-dose), but the simple comparison between the two regimens suggests that primaquine has a better performance. If this is the reality, it should be clearer from the text, as well as a reference to the final cost-benefit balance, more favourable to Tafenoquine.

We thank the reviewer for this comment. This is an important point that merits more discussion and clarity in the text. The odds-ratios are for 1mg/kg increments for each drug. A 1mg/kg increment in the total primaquine dose is of course not strictly comparable to a 1mg/kg increment in the total tafenoquine dose. However, if we assume that 5mg/kg of tafenoquine is equivalent to 3.5mg/kg total dose primaquine (they are at least equivalent in terms of current dosing recommendations), then, expressed as percentage increases, a 1 mg/kg increment in the tafenoquine dose (20% increase) would be equivalent to a 0.7mg/kg increment in the total primaquine dose. The odds-ratio for recurrence at 4 months with a 0.7mg/kg increment in the primaquine dose is 0.72 (close to identical to the tafenoquine odds-ratio).

We have revised the text to highlight this important point (lines 125-134):

“Under the model fit to all data (n=1,073), each additional mg/kg of tafenoquine was associated with an odds-ratio (OR) for recurrence within 4 months of 0.70 (95% CI: 0.65 to 0.76). A 1mg/kg tafenoquine increase corresponds to a 20% increase relative to the currently recommended dose of 5mg/kg (assuming a patient weight of 60kg). In comparison, a 20% increase in the total primaquine dose corresponds to 0.7mg/kg for a target dose of 3.5mg/kg. This was associated with an odds ratio for recurrence within 4 months of 0.72 (95% CI 0.66 to 0.78), i.e. a very similar dose effect for the two drugs. Our model estimated that the mean probability of recurrence at 4 months was 0.15 (95% CI: 0.10 to 0.23) following 5mg/kg tafenoquine and 0.17 (95% CI: 0.10 to 0.26) following 3.5mg/kg primaquine. The posterior probability that the recurrence was more likely following primaquine was 0.73.”

"MetHb concentration was associated with an odds ratio for recurrence of 0.81 (95% CI 0.65 to 0.99). Furthermore, consistent with this association between MetHb and vivax malaria recurrence (Figure 4), the day 7 MetHb levels were weakly inversely correlated with the tafenoquine weight-adjusted t1_2 values (Figure 3d; ⇾ = *0.1, p=0.05)."Production of methemoglobin is associated with the therapeutic effects of tafenoquine (and primaquine). This might be related to the fact that the tafenoquine (and primaquine) clinically active metabolites that kill the parasite are also involved in oxidative stress creating higher levels of methemoglobin. In this scenario, I suggest that the authors investigate associations between the available CYP2D6 status and D7 methemoglobin levels. Metabolism affects the T_1/2_ of a drug, and knowing this PK factor is associated with increased efficacy, the hypothesis of discerning a CYP2D6 influence is in my view interesting. Further, this information should be incorporated into the presented PD modelling. In a nutshell, are different CYP2D6 allele statuses related to different curves of incremented efficacy (OR ~ δ Mg/Kg)? Are there human populations "promoting" a better tafenoquine performance, or is this associated with increased adverse events affecting the final cost-benefit of increasing the dosing?In this context, I regret the lack of information concerning CYP2D6 copy number variation, a mutation relatively common in SE Asia. This is an obvious consequence of the limitations of retrospective studies.Finally, it is to note that CYP2D6 activity scores are derived from experimentation using specific probe drugs. Albeit generally informative, these scores (a quite rough metric on their own) might not reflect the enzyme capacities when considering other substrates, such as the 8-aminoquinolines. Also, the data presented, especially when concerning CYP2D6 has to be considered partial. This gene is highly variable, and only a full ORF-phased sequencing can establish its true primary sequence nature (e.g. see Gaedick et al., 2018, J Pers Med. 2018 Apr 17;8(2):15. doi: 10.3390/jpm8020015)

We agree that human genetic polymorphisms in CYP2D6 are potentially important, however they are complex ranging from rare gain of function gene duplications to homozygous or mixed heterozygote mutations conferring varying degrees of loss of function. Allele frequencies vary markedly in the human population, but the most important in terms of prevalence in populations at risk from *P. vivax* infection is the *10 loss of function mutation which reaches allele frequencies of up to 40% in populations genetically related to the Han Chinese. Unfortunately, although our analysis is large (651 patients received tafenoquine), even for *10 the number patients with identified polymorphisms are too small for confident assessment.

An analysis looking at the relationship between CYP2D6 and the terminal half-life is of course restricted to patients who received tafenoquine. Out of the 651 patients treated with tafenoquine, only 391 were genotyped for CYP2D6 mutations and had sufficient drug levels to estimate a terminal half-life. Of these 391 patients, only 35 had an activity score of 0.5 or less. This would only provide statistical power to detect very large changes in terminal half-life or recurrence probability. Hence whilst we agree that the role of CYP2D6 is very important our series is underpowered to provide confident answers.

To clarify this we have revised the results and added an exploratory analysis of the relationship between tafenoquine elimination half-life and CYP2D6 activity score:

Lines 234-236 (Results): “In addition, there was no clear association between having an activity score ≤0.5 (n=35 vs n=356) and tafenoquine elimination half-life (-0.15 days change in the poor versus normal/extensive metabolisers, 95% CI: -1.22 to 0.94).”

Lines 351-357 (Limitations paragraph in Discussion): “A final limitation concerns the role of CYP2D6 human genetic polymorphisms in the metabolism of primaquine. Only two thirds of enrolled patients were genotyped for CYP2D6 mutations. In patients who received tafenoquine, only 35 were poor metabolisers. This makes it difficult to say with certainty what the role CYP2D6 has in mediating tafenoquine metabolism. However, it highlights the fact that in these vivax malaria endemic areas, even if CYP2D6 polymorphisms did play a key role, relatively few patients are null metabolisers. Thus CYP2D6 is a minor determinant of efficacy at the population level compared with drug dosing.”

"The phase 1 study was a drug-drug interaction study, with volunteers randomised to either tafenoquine alone or tafenoquine in combination with DHA-PQP or artemether-lumefantrine (AL), the standard antimalarial treatment regimens in both cases."Lumefantrine is well known to be a competent CYP2D6 inhibitor. I did not find any indication in the report of this factor being explored and incorporated into the overall model. Can the authors elaborate on the data concerning AL-treated subgroups?

The degree to which this theoretical interaction affects tafenoquines radical curative activity is unknown. A priori we thought it would not be strong enough to influence methaemoglobin production. However, when comparing day 7 lumefantrine levels with day 7 methaemoglobin concentrations in healthy volunteers, there was a suggestion that higher lumefantrine exposure was associated with lower methaemoglobin production.

There were 23 volunteers with a day 7 lumefantrine level and a day 7 methaemoglobin measurement. A ten-fold increase in day 7 plasma lumefantrine concentration was associated with a 40% reduction in day 7 methaemoglobin (95% CI: 3 to 62). In comparison, the relationship between a ten-fold increase in day 7 piperaquine concentration and day 7 methaemoglobin was not significant (40%; 95% CI: -27 to 72). This data set is small however and these results need further confirmation.

We have added to the Results on lines 215-221:

“Lumefantrine may inhibit cytochrome P450 2D6 (CYP2D6) (Novartis Pharmaceuticals, 2021, see next section). We compared day 7 lumefantrine concentrations and day 7 piperaquine concentrations with the day 7 MetHb concentration in individuals randomised to tafenoquine + ACT in the phase 1 trial (Green et al., 2016). Higher plasma lumefantrine concentrations were associated with lower day 7 MetHb levels (ten-fold increase in day 7 plasma lumefantrine concentration was associated with a 40% reduction in day 7 methaemoglobin (95% CI: 3 to 62), Supplementary Figure 14). In contrast, for piperaquine there was no clear association with the day 7 MetHb concentrations (40%; 95% CI: -27 to 72).”

Reviewer #2 (Recommendations for the authors):Congratulations on the paper.

We thank the reviewer for their kind and enthusiastic comments.

The sensitive analysis for the 4 vs 6 months outcome could be further emphasized, as shortening the time to the last outcome can be very useful for future clinical trials.

We have added to the Discussion on lines 326-331:

“We note that the similar results of 4-month and 6-month efficacy endpoints suggests that future antirelapse efficacy studies could be restricted to a follow up of four months. This has substantial implications for the cost and operational feasibility of future antirelapse studies. Not only did most observed recurrences occurred within 4 months follow up (83%, 298 out of 360), but this endpoint also had considerably less loss to follow-up.”

It would be interesting to have a prediction of the haemolysis in case of the unfortunate inadvertent use of the new dose in G6PD deficient female heterozygotes. This would be the primary clinical concern in remote areas.

We agree. Whether the higher dose would result in greater haemolysis (as would be expected from primaquine) or whether the haemolytic effect is already largely saturated at the 300mg dose is not known. We have no data on which to base a prediction!

The conclusion states: Patient weight was not a predictor of recurrence per se. This could be clarified.

We have the following sentence on line 346:

“However, patient weight was not associated with recurrence.”